# Harmonized chronologies of a global late Quaternary pollen dataset (LegacyAge 1.0)

Chenzhi Li[1,2], Alexander K. Postl[1], Thomas Böhmer[1], Xianyong Cao[1,3], Andrew M. Dolman[1], Ulrike Herzschuh[1,2,4]

[1] Alfred Wegener Institute, Helmholtz Centre for Polar and Marine Research, Polar Terrestrial Environmental Systems, Telegrafenberg A45, 14473 Potsdam, Germany

[2] Institute of Environmental Science and Geography, University of Potsdam, Karl-Liebknecht-Str. 24-25, 14476 Potsdam, Germany

[3] Alpine Paleoecology and Human Adaptation Group (ALPHA), State Key Laboratory of Tibetan Plateau Earth System, and Resources and Environment (TPESRE), Institute of Tibetan Plateau Research, Chinese Academy of Sciences, 100101 Beijing, China

[4] Institute of Biochemistry and Biology, University of Potsdam, Karl-Liebknecht-Str. 24-25, 14476 Potsdam, Germany

**Correspondence:** Ulrike Herzschuh (Ulrike.Herzschuh@awi.de)

**Abstract.** We present a chronology framework named LegacyAge 1.0 containing harmonized chronologies for 2831 pollen records (downloaded from the Neotoma Paleoecology Database and the supplementary Asian datasets) together with their age control points and metadata in machine-readable data formats. All chronologies use the Bayesian framework implemented in Bacon version 2.5.3. Optimal parameter settings of priors (accumulation.shape, memory.strength, memory.mean, accumulation.rate, thickness) were identified based on information in the original publication or iteratively after preliminary model inspection. The most common control points for the chronologies are radiocarbon dates (86.1%), calibrated by the latest calibration curves (IntCal20 and SHcal20 for the terrestrial radiocarbon dates in the northern and southern hemispheres; Marine20 for marine materials). The original publications were consulted when dealing with outliers and inconsistencies. Several major challenges when setting up the chronologies included the waterline issue (18.8% of records), reservoir effect (4.9%), and sediment deposition discontinuity (4.4%). Finally, we numerically compare the LegacyAge 1.0

chronologies to those published in the original publications and show that the reliability of the chronologies of
95.4% of records could be improved according to our assessment. Our chronology framework and revised
chronologies provide the opportunity to make use of the ages and age uncertainties in synthesis studies of, for
example, pollen-based vegetation and climate change. The LegacyAge 1.0 dataset, including metadata, datings,
harmonized chronologies, and R code used, are open-access and available at PANGAEA
(https://doi.pangaea.de/10.1594/PANGAEA.933132;  Li  et  al.,  2021)  and  Zenodo
(https://doi.org/10.5281/zenodo.5815192; Li et al., 2022), respectively.

**1 Introduction**
Global and continental fossil pollen databases are used for a variety of paleoenvironmental studies, such as past
climate and biome reconstructions, palaeo-model validation, and the assessment of human-environmental
interactions (Gajewski, 2008; Gaillard et al., 2010; Cao et al., 2013; Mauri et al., 2015; Trondman et al., 2015;
Marsicek et al., 2018; Herzschuh et al., 2019). Several fossil pollen databases have been successfully established
(Gajewski,  2008;  Fyfe  et  al.,  2009),  such  as  the  European  Pollen  Database
(http://www.europeanpollendatabase.net),  the  North  American  Pollen  Database
(http://www.ncdc.noaa.gov/paleo/napd.html),  and  the  Latin  American  Pollen  Database
(http://www.latinamericapollendb.com); most of these data are now included in the Neotoma Paleoecology
Database (https://www.neotomadb.org/; Williams et al., 2018). Chronologies and age control points are stored in
these databases along with the pollen records.
However, to date, the metadata and dating results of these records are not available in a machine-readable
format; furthermore, the chronologies have been established using a variety of methodologies, and the
quantification of temporal uncertainty, particularly between records, remains a challenge (Blois et al., 2011;
Giesecke et al., 2014; Flantua et al., 2016; Trachsel and Telford, 2017). Recently, the need for harmonized and
consistent chronologies allowing for the accurate assessment of temporal uncertainty between records has
increased as studies are looking for spatiotemporal patterns using multi-record analyses (Jennerjahn et al., 2004;
Blaauw et al., 2007; Giesecke et al., 2011; Flantua et al., 2016). Accordingly, some effort has been made to
harmonize the chronologies for a subset of the records in these databases (Fyfe et al., 2009; Blois et al., 2011;
Giesecke et al., 2011; Giesecke et al., 2014; Flantua et al., 2016; Brewer et al., 2017; Wang et al., 2019; Mottl et
al., 2021). However, a harmonized chronology framework is needed, not only to allow for the consistent inference
of age and age uncertainties but also to apply to newly published records or one that can be adjusted to the specific
requirement of a study.
Here we present the rationale and code, as well as the metadata and parameter settings for the chronology
framework LegacyAge 1.0, which contains harmonized chronologies for 2831 palynological records, synthesized
from the Neotoma Paleoecology Database (last access: April 2021, Neotoma hereafter) and the supplementary
Asian datasets (Cao et al., 2013, 2020). We also report on the major challenges of setting up the chronologies and
assessing their quality. Finally, the newly harmonized chronologies are numerically compared with the original
ones. All data and R code used for this study are open-access and available at PANGAEA
(https://doi.pangaea.de/10.1594/PANGAEA.933132;     Li     et     al.,     2021)     and     Zenodo
(https://doi.org/10.5281/zenodo.5815192; Li et al., 2022), respectively.

**2 Methods**
**2.1 Data sources**
We established harmonized chronologies for 3471 records in the 'Global taxonomically harmonized late
Quaternary pollen dataset' (https://doi.pangaea.de/10.1594/PANGAEA.929773; Herzschuh et al., 2021). This
compilation comprises 3147 records from Neotoma (last access: April 2021) and 324 Asian records from China
and Siberia compiled by Cao et al. (2013, 2020) and from our own data (AWI). Records are from lake sediments
(49.4%), peatlands (34.3%), and other archives (16.3%) (Fig. 1). The following chronology metadata were
collected for each record: *Event, Data_Source, Site_ID, Dataset_ID, Site_name, Location (longitude, latitude,*
*elevation, and continent), Archive_Type, Site_Description, Reference, Laboratory_label, Dating_Method,*
*Material_Dated, Date (uncalibrated and calibrated age, error older, error younger, depth, thickness), Additional*
*relevant comments from authors (e.g., reservoir effect, hiatus, outliers, and date rejected).* Furthermore,
information on the original chronologies of each pollen record was also taken from the Neotoma and
supplementary Asian datasets, including *Chronology_name, Age_type (calibrated or uncalibrated radiocarbon*
*years BP), Pollen_depth, Estimated age (age, age error)).* These metadata are available at
https://doi.pangaea.de/10.1594/PANGAEA.933132 (Supplement Table S1 and S4; Li et al., 2021).

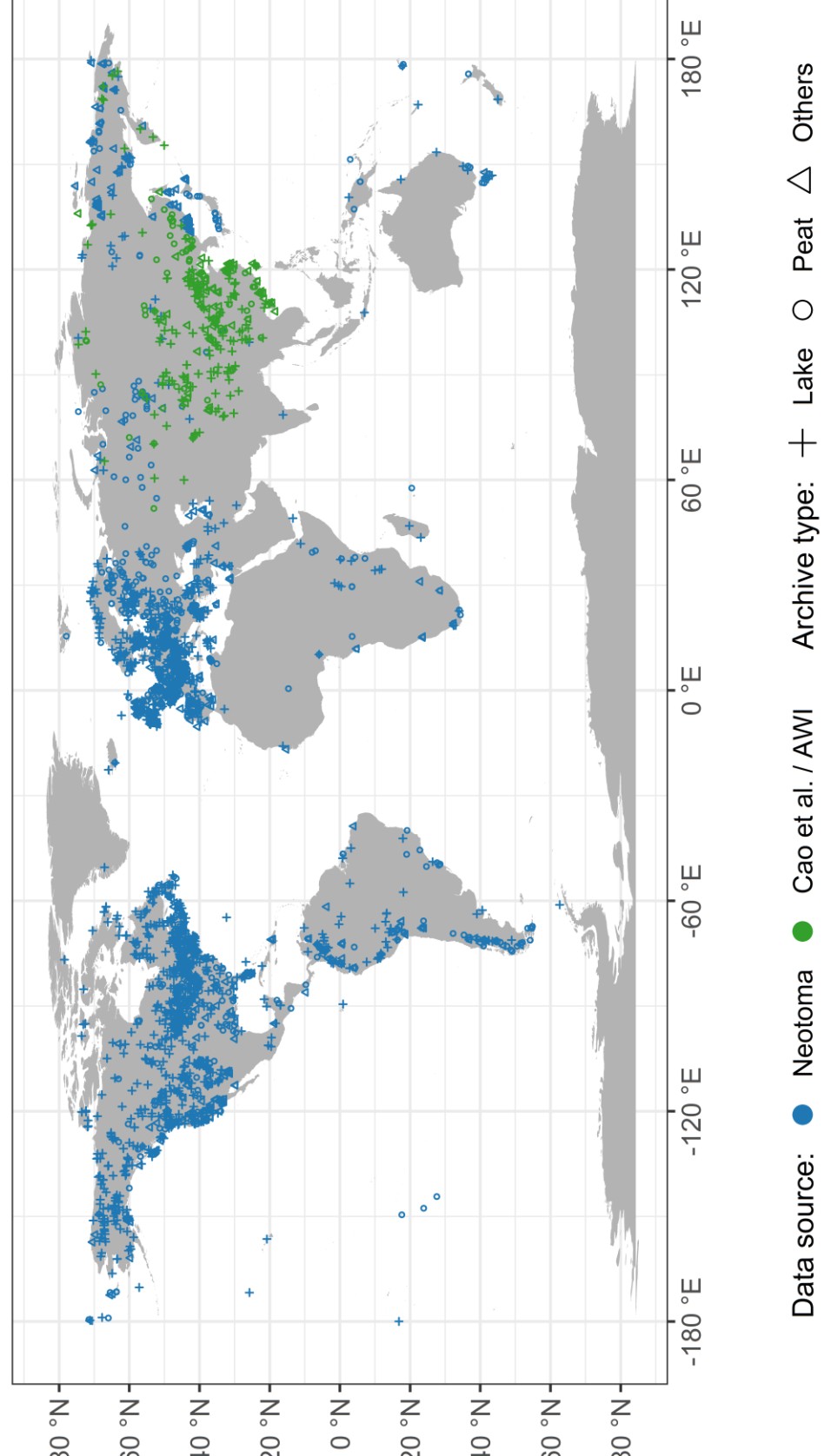

**Figure 1.** Map of records by source and archive type.

**2.2 Chronological control points**
**2.2.1 Radiometric dates**
***Radiocarbon dating:*** most records were dated using radiocarbon-based methods ($^{14}C$ dating, conventional or
accelerator mass spectrometry, Christie, 2018), coverimg the time range of ca. the last 50 kyr BP (before present,
where 'present' is 1950 CE). However, the accuracy and precision of the radiocarbon dates depend on the
calibration curve, taphonomy, and dating materials (Blois et al., 2011; Heaton et al., 2021).
***Lead-210 dating:*** the uppermost part of some lake records has been dated using a radioactive isotope of lead
(lead-210), which has a half-life of ca. 22 years and provides useful age control for the last 75-150 years. However,
the abundance of other radioactive isotopes (e.g., Caesium-137) affects the accuracy and precision of the
calibration curve for lead-210, resulting in temporal uncertainty (Appleby and Oldfield, 1978; Cuney, 2021).
***Luminescence dating:*** archaeological materials, loess, and river sediments have often been dated via
luminescence, including thermoluminescence (TL) and optically stimulated luminescence (OSL), which cover
time scales from millennia to hundreds of thousands of years (Roberts, 2013). Due to the systematic and random
errors in the measurement process, the luminescence ages have at least 4-5% uncertainty, which widens with
increasing time (Wallinga and Cunningham, 2015).
**2.2.2 Lithological dates**
***Varve dating:*** varve chronology, generated from counting varves, is considered a relatively accurate dating
method for the late Quaternary, particularly the Holocene. Although sediment characteristics (e.g., thickness,
continuity, marking layer) may create uncertainty in varve-counted ages, these uncertainties are small relative to
those from radiometric methods (Ojala et al., 2012; Zolitschka et al., 2015; Ramisch et al., 2020). If a pollen
record has a varve chronology stored and assessed in the Varved Sediments Database (VARDA, https://varve.gfz-
potsdam.de/), we generally prefer to use it over chronologies based on other dating techniques.
***Tephrochronology:*** tephra layers are used as isochrones to correlate and synchronize sequences at a regional or
continental scale (Lowe, 2011). The uncertainties of tephrochronology are similar to those known in radiocarbon
dating, such as methodological and dating errors (Flantua et al., 2016). Tephras documented in the Global
Tephrochronological Database (Tephrabase, https://www.tephrabase.org/) were included to improve the
chronologies, such as the Mazama ash (7630+-40 cal. yr BP; Brown and Hebda, 2003), Vedde ash (12121+-57
cal. yr BP; Lane et al., 2012), and the Laacher See ash (12880+-120 cal. yr BP).

### 2.2.3 Biostratigraphical dates

Biostratigraphical dates have been widely relied on before [14]C dating became available and affordable (Bardossy
and Fodor, 2013). We ignored most of the available biostratigraphical dates when we harmonized the chronologies
because vegetation reaction to climate change is likely not sufficient synchron. Only a few well-known and widely
applicable biostratigraphic boundaries (Rasmussen et al., 2014) were used in other dating techniques that could
not sufficiently constrain the chronologies, for example, the Younger Dryas/Holocene (11500±250 cal. yr BP),
Allerød/Younger Dryas (12650±250 cal. yr BP), and Oldest Dryas/Bølling (14650±250 cal. yr BP; Giesecke et
al., 2014).

### 2.3 Establishing the chronologies

### 2.3.1 Method choice

We used the Bacon software (Blaauw and Christen, 2011) to establish continuous down-core chronologies from
the age control points. Bacon fits a monotonic autoregressive (AR1) model to age control points using Bayesian
methods to combine information from the control points with prior information on the statistical properties of
accumulation histories for deposits, e.g., a prior distribution for the mean accumulation rate and how it varies
(Blaauw and Christen, 2011). Several other approaches are available for age-depth modeling, including linear
interpolation, smoothing splines, and other Bayesian methods, e.g., OxCal (Ramsey, 2008) and Bchron (Haslett
and Parnell, 2008). However, Bacon has become one of the most frequently used and compares well with other
methods (Trachsel and Telford, 2017, Blaauw et al., 2018).
Bacon provides the calibrated ages (mean, median, minimum, maximum) at each depth (e.g., every centimeter)
with a 95% confidence intervals and an indication of how well the model fits the dates, although it needs much
supervision and computing power. The prior distribution guides the overall trend of the age-depth relationships,
so the control points guide rather than strictly constrain the age-depth relationships (Giesecke et al., 2014). Bacon
version 2.3.3 and later (Blaauw and Christen, 2011) can also handle sudden shifts in the accumulation rate when
given the hiatus/boundary depth and resetting the memory to 0 when crossing the hiatus. Therefore, all age-depth
relationships in our dataset will be constructed using the latest Bacon version 2.5.3 (Blaauw and Christen, 2011;
Blaauw et al., 2018) in R (R Core Team, 2021).

### 2.3.2 Core tops and basal ages

Wherever possible, the record-related publications were read to decide whether the core top was modern at the
time of sampling. For modern core-tops, if the core was collected from sites where sediment was still accumulating,
the sediment surface could be assigned to the year of sampling, adding one significant time control for the
chronologies. If the sampling date was unavailable, an alternative surface age from the original chronology in
Neotoma was added at the core top. An estimated artificial core-top age (-50 + -30 cal yr BP) was used if none of
the above ages were available (Supplement Table S2, S3). We inferred the surface age from the calibrated age-
depth model for core-tops judged not to be modern. For basal ages, when the calibrated age-depth model for the
lowermost profile has considerable extrapolation and was not sufficiently constrained by the control points, we
also accepted the prior information of core basal age from the record-related publications or Neotoma.

### 2.3.3 Calibration curves

To transform the measured $^{14}$C ages to calendar ages, the latest calibration curves, approved by the radiocarbon
community (Hajdas, 2014), were used in Bacon routine: IntCal20 (Reimer et al., 2020; Heaton et al., 2021) and
SHcal20 (Hogg et al., 2020) to calibrate the terrestrial radiocarbon dates in the northern and southern hemispheres,
respectively; and Marine20 (Heaton et al., 2020) for the 38 marine records included in our dataset (Sánchez Goñi
et al., 2017). The numerical probability distributions of calendar age from calibrated radiocarbon dates were
summarised to a mean and standard deviation for use in Bacon. Absolute dates (e.g., lead-210, OSL, tephra),
already presented on the calendar scale, were not calibrated (Blaauw and Christen, 2011). Modern/post-bomb $^{14}$C
dates (negative $^{14}$C ages) were calibrated using appropriate post-bomb calibration curves (post-bomb=1 for
>40°N; 2 for 0°-40°N; 4 for southern hemisphere; Hua et al., 2013).

### 2.3.4 Parameter settings for the initial Bacon run

After consultation of the relevant publication (Blaauw and Christen, 2011; Goring et al., 2012; Cao et al., 2013;
Fiałkiewicz-kozieł et al., 2014; Blaauw et al., 2018) and assessments of several runs with a test set of records, we
set the following Bacon parameters (Supplement Table S3):
(1) The prior for the accumulation rate consists of a gamma distribution with two parameters, **mean accumulation**
**rate** (acc.mean; default 20 yr cm$^{-1}$) and **accumulation shape** (acc.shape; default 1.5). For the acc.shape, we
accepted its default value as higher values resulted in a more peaked shape of the gamma distribution. A first
approximation of the acc.mean was calculated as the average accumulation rate between the first and the last
date of each record, combined with the prior information of dates, which is more reasonable than using a
constant value.
(2) Bacon divides a core into many vertical sections of equal **thickness** (thick; default 5 cm), which significantly
affects the flexibility of the age-depth model, and through millions of Markov Chain Monte Carlo iterations
estimates the accumulation rate for each section. Blaauw and Christen (2011) indicated that models with few
sections tend to show more abrupt changes in accumulation rate, while models with many sections usually
appear smoother but are computationally more intense. We run Bacon for six section thicknesses (2.5 cm, 5
cm, 10 cm, 30 sections, 60 sections, and 120 sections), optimal values after numerous tests, with and without
core-top age resulting in 12 initial chronologies for each record.
(3) The prior for the memory, that is, the dependence of accumulation rate between neighboring depths, is a beta
distribution defined by two parameters: **memory strength** (mem.strength; default 10) and **mean memory**
(mem.mean; default 0.5). For the mem.strength, we used a value of 20 as suggested by Goring et al. (2012),
which allows a large range of posterior memory values. We set different mem.mean values (0.3 for lake and
0.7 for peatland) to accommodate differences in accumulation conditions between lakes and peatland, where
the higher memory for peatlands implies a more constant accumulation history (Blaauw and Christen, 2011;
Goring et al., 2012; Cao et al., 2013; Cao et al., 2020).
(4) The **minimum (maximum) depth** (d.min and d.max, respectively) of the age-depth model was defined by the
uppermost (lowermost) dating or pollen sample depth (Supplement Table S4). The parameter 'd.by' (default
1 cm) defines the **depth intervals** at which ages are calculated, and we accepted its default value.
In addition to the major parameters mentioned above, we also adjusted several additional parameters for
individual records according to prior information collected from record-related publications or Neotoma
(Supplement Table S2, S3).
**(1) Reservoir effects:** the uptake of old carbon by aquatic plants, mosses, or shells either originating from, e.g.,

limestone in the catchment ('hard-water effect') or slow $^{14}$C exchange between the atmosphere and ocean

interior, can result in too old radiocarbon dates (Philippsen, 2013; Philippsen and Heinemeier, 2013; Giesecke

et al., 2014; Heaton et al., 2020). In addition to the reservoir ages reported by the original authors, we also

identified some additional records for which there is likely a reservoir effect through modern correction and

linear extrapolation (Wang et al., 2017). We then subtracted the reservoir age as a constant from all $^{14}$C dates

of an affected record, excluding those derived from terrestrial macrofossils. We may have underestimated the

number of such records due to the difficulty of estimating the reservoir age where the sediment surface was

eroded or used for agricultural purposes.

**(2) Waterline issues:** stratigraphic records do not always start at a depth of 0 cm, for example, if the uppermost

part of the core is lost, if the record is only a part of a longer sequence, or if the depths are measured from the

water surface instead of the sediment surface, leading to the so-called waterline issue. Accordingly, we

adjusted the uppermost depth of the chronology based on information collected from the original publications

and Neotoma.

**(3) Hiatuses:** where sediment deposition was not continuous, it is possible to set a "hiatus" at which Bacon resets

the memory to 0, causing a break in the autocorrelation in the accumulation rate for depths before and after

the hiatus and additionally models an instantaneous jump in age at that depth (Blaauw and Christen, 2011).

**(4) Dates rejected/added:** Neotoma usually reports all $^{14}$C dates from cores, even when deemed inaccurate. We

assessed prior information on dates and then excluded the $^{14}$C dates of samples with contaminated or reworked

sediments from age-depth model from age-depth models, in most cases following the suggestions in the

original publications. For example, we excluded the date at 164 cm, accepted by the author (Gajewski et al.,

2000), from the *Muskoka Lake* record (ID 1783), as it does not agree with the other three dates from the same

core and where lithology had changed significantly at that depth. We down-weighted the impact of outliers on

the overall trend of the age-depth relationships and risked that age uncertainties were too optimistic. We also

documented all lithological dates (e.g., varves and tephra) and biostratigraphical dates collected from the

original publications and Neotoma to supplement the chronology metadata.

**2.3.5 Assessment of initial age-depth models and final parameter selection**
To objectively evaluate the 12 initial age-depth models for each record, we initially tested a least-squares method
between the age model and ages of dated depths and calculated the mean uncertainty for each model. However,
the least-squares method is susceptible to outliers (Birks et al., 2012), and models with least-squares may risk
more abrupt changes in accumulation rate due to over-fitting dates. Instead of a numerical comparison, we finally
implemented a visual comparison based on the Bacon output graphs, which show the Markov Chain Monte Carlo
iterations, the prior and posterior distributions for the accumulation rate and memory, and how well the model fits
the date (Blaauw and Christen, 2011).
Preference was given to models that fitted the dates well, had small mean uncertainties (Supplement Table S5),
and good runs of Markov Chain Monte Carlo iterations (i.e., a stationary distribution with little structure among
neighboring iterations as indicated by the traceplot of the joint likelihood) when visual choosing the 'best' model
for each record (Blaauw and Christen, 2011; Blaauw et al., 2018). If necessary, we adjusted the parameter settings
such as the section thickness and mean accumulation rate to better fit with the dates that were consistent with prior
information. For the final parameter settings used for each record, please see
https://doi.pangaea.de/10.1594/PANGAEA.933132 (Supplement Table S3; Li et al., 2021).

**2.4 Evaluation of the newly generated age-depth models**
For the temporal uncertainty of the age-depth models, we take used the 95% confidence intervals for age estimated
by the Bacon model for each centimeter (Supplement Table S5). These values are approximately twice the
standard error of the estimated age at a given depth. We plotted our newly generated 'best' calibrated chronologies
with 95% confidence intervals together with the original ones taken from the Neotoma and Cao et al. (2013, 2020)
datasets (Supplement Table S4) to compare and evaluate the performance of the new models visually. The criteria
for the preferred models are that the model fitted the dates well, had small uncertainties, combined dates with
prior information (e.g., geological and hydrological setting, environmental history), and calibrated with the latest
calibration curves.
**3 Results**
**3.1 Overview of major challenges when establishing the chronologies**
Age-depth models were initially established for all 3471 records in the harmonized pollen data collection
(Herzschuh et al., 2021). We discarded 640 records with fewer than two reliable dates (i.e., no reliable date or
only one reliable date), evaluated based on prior information from original literature, leaving chronologies for
2831 records. We faced several major challenges when establishing the chronologies. After assessments and
consultation of prior information from original publications (Supplement Table S2, S3), we identified 139 records
(4.9%) with reservoir effects, 533 records (18.8%) with waterline issues, 125 records (4.4%) with hiatuses, 924
records (32.6%) with rejected or added dates, and 743 records (26.2%) that contained several of the above
problems: all these challenges have been handled (Fig. 2). After assessing initial age-depth models, accumulation
rates were adjusted for 367 records (13.0%), and different section thicknesses were applied to 411 records (14.5%).

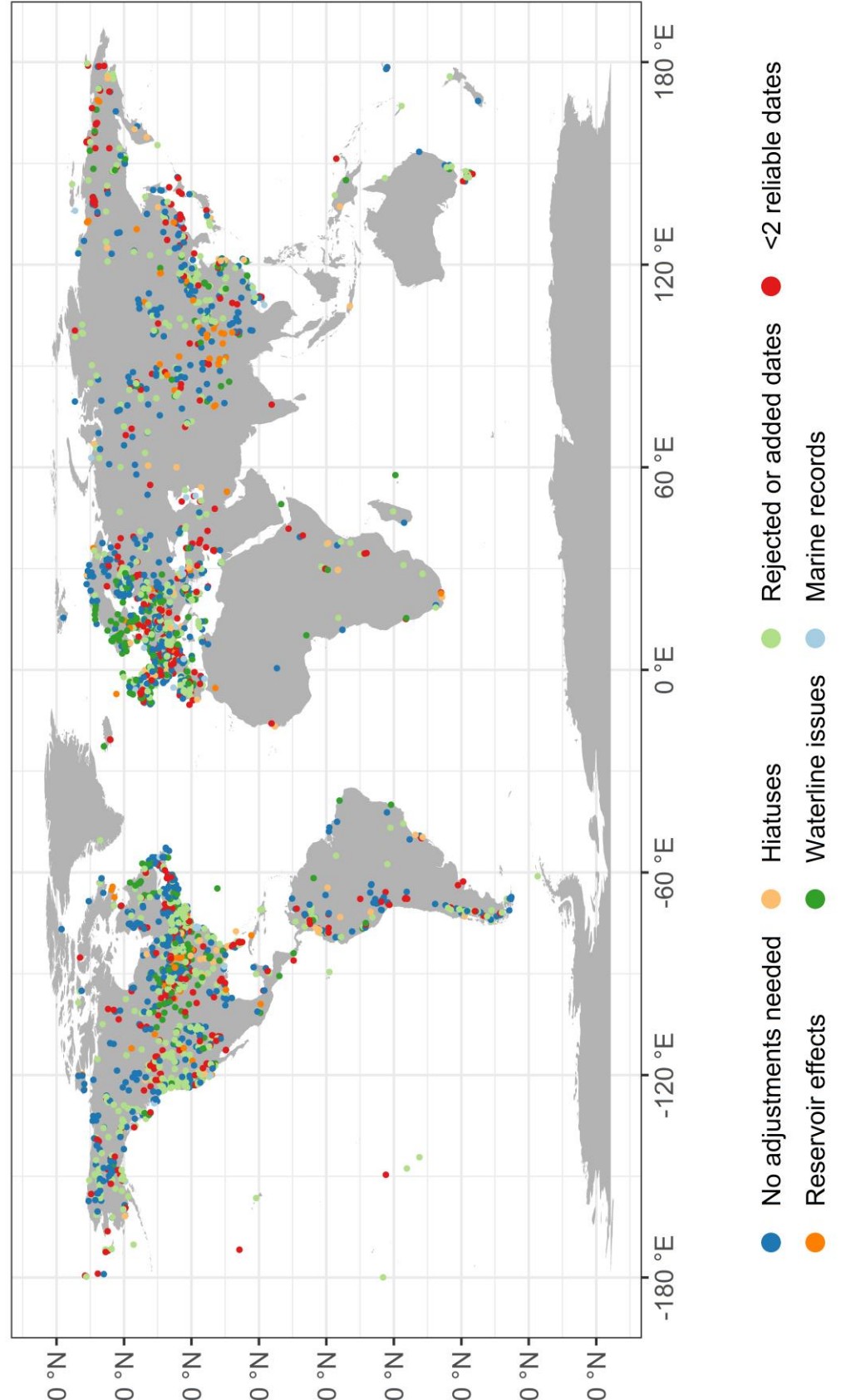

**Figure 2.** The distribution of records that faced various major challenges when establishing their chronologies.


**3.2 LegacyAge 1.0 quality**
**3.2.1 Dates used for final chronologies**
A total of 19,990 control points (out of 21,199 dates available) were used to generate the chronologies for the
2831 records (Supplement Table S1). Among them, the most common chronological control points are
radiocarbon dates (86.1%), followed by lithological and biostratigraphical dates (8.5%) collected from
publications or Neotoma, and lead-210 (5.0%); other dating techniques make up 0.4% of the control points. The
median number of dates per chronology is 5, with 23.3% of the chronologies having 2 or 3 dates, 53.3% having
4-8 dates, and 23.4% having at least 9 dates (Fig. 3).

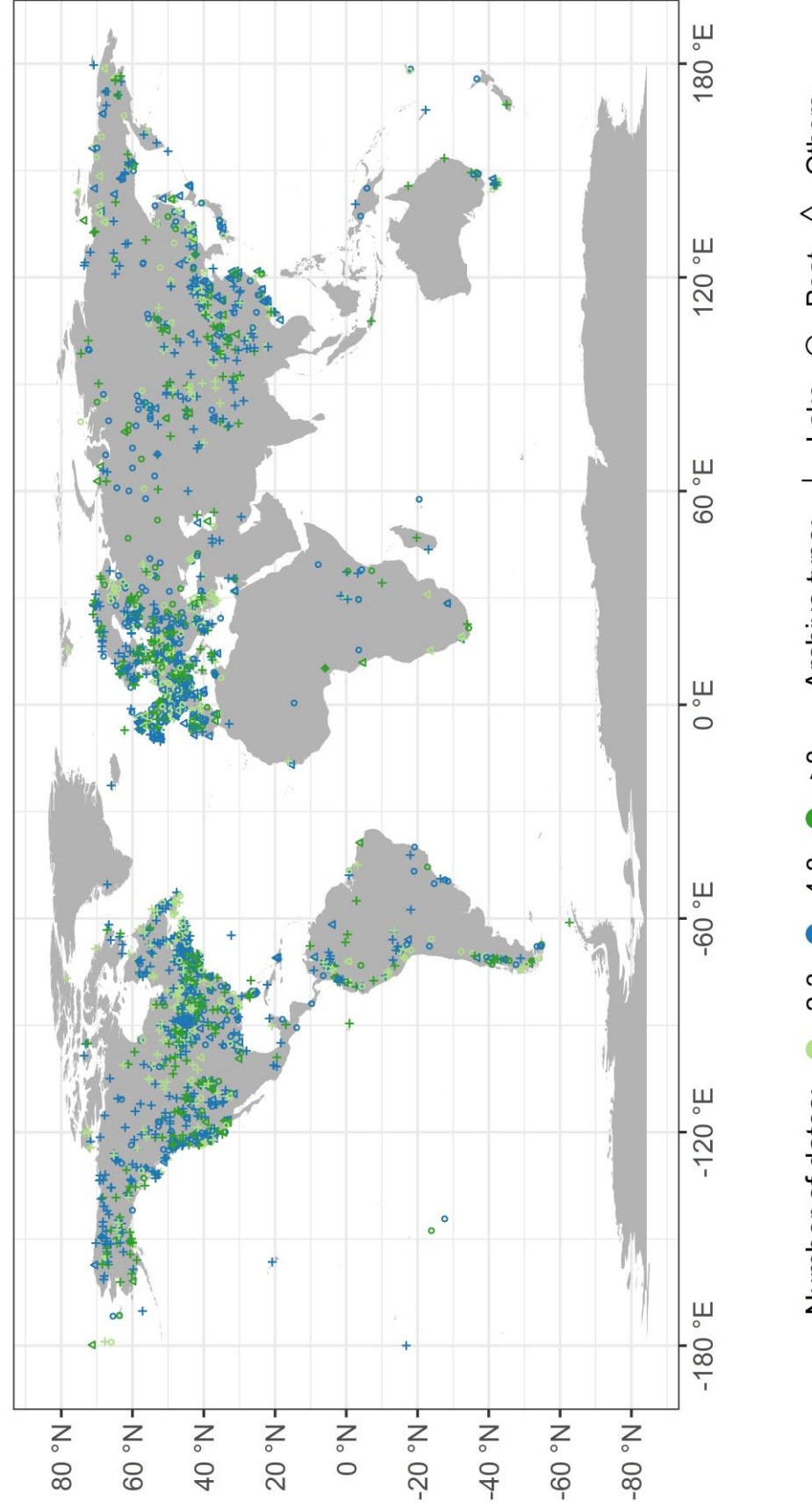

**Figure 3.** Map of the number of dates and archive types for each record.


Currently, 80.5% of chronological control points in the LegacyAge 1.0 fall within the Holocene (37.9%, 25.2%,
and 17.4% within the late (ca. 0-4.2 cal. kyr BP), middle (ca. 4.2-8.2 cal. kyr BP), and early Holocene (ca. 8.2-
11.7 cal. kyr BP), respectively), 14.5% within the Last Deglaciation (ca. 11.7-19.0 cal. kyr BP; Clark et al., 2012),
2.0% within the Last Glacial Maximum (LGM; ca. 19.0-26.5 cal. kyr BP; Clark et al., 2009), and only 3.0% earlier
than the LGM (Fig. 4).

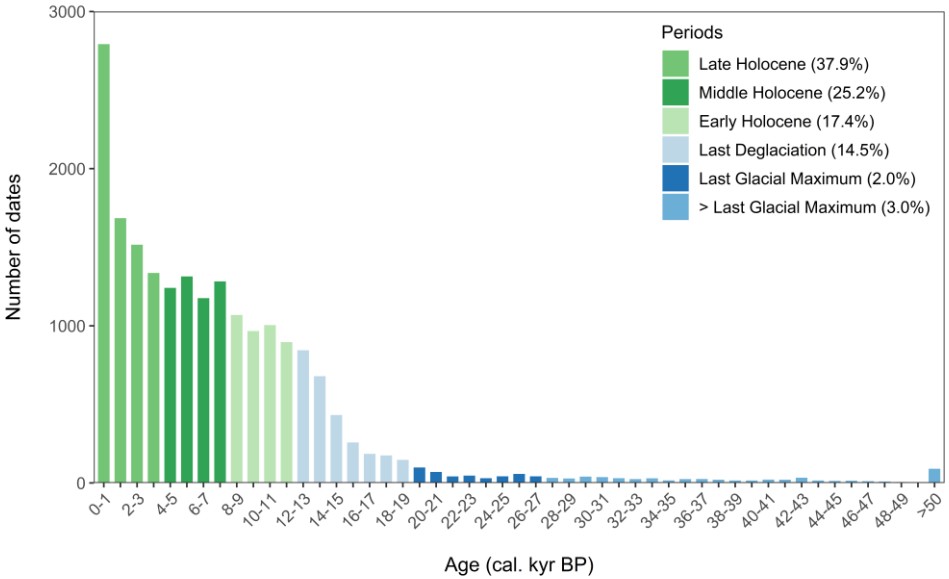


**Figure 4.** Histogram showing the number of available dates in distinct time slices.

**3.2.2 Spatial and temporal coverage**
Of the 2831 chronologies finally established, 1032 records are from North America, 1075 records from Europe,
488 records from Asia, 150 records from South America, 54 records from Africa, and 32 records from the Indo-
Pacific (Fig. 3). Most records (2659 records, 93.9%) are in the northern hemisphere, where the main vegetation
and climate zones are covered.
As shown in Fig. 5, 94.8% of chronologies cover part of the last 30 kyr, while Marine Isotope Stage 3 (MIS-3)
is relatively poorly covered. Specifically, 98.0% of chronologies cover part of the Holocene (90.7%, 81.0%, and
65.8% cover part of the late, middle, and early Holocene, respectively), 46.7% cover part of the Last Deglaciation,
10.7% cover part of the Last Glacial Maximum, and only 6.1% earlier than LGM.

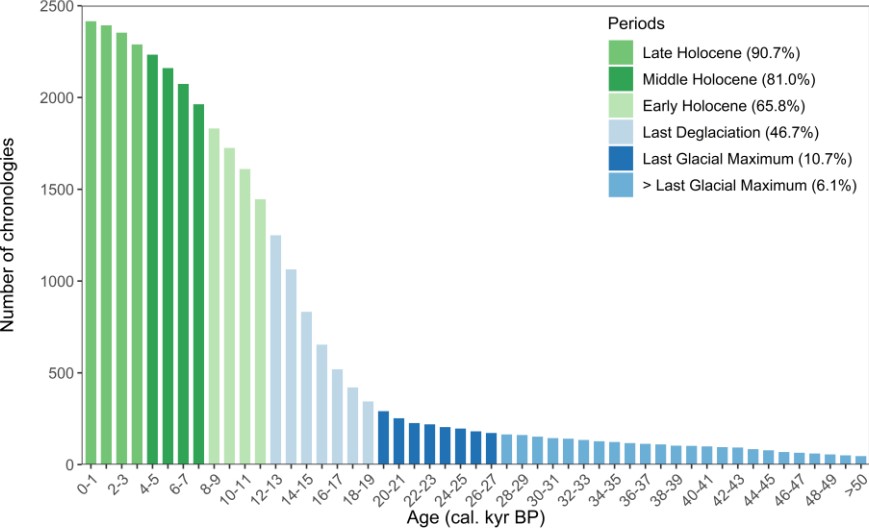

**Figure 5.** Histogram showing the number of available chronologies in distinct time slices.

### 3.2.3 Temporal uncertainty

Boxplots of age uncertainties for all chronologies in distinct time slices (Fig. 6), excluding outliers (ca. 5.1%), illustrate that age uncertainty tends to increase with age and is mainly related to the uncertainty and precision of the chronological control points, calibration curves, and age models (Blois et al., 2011). The boxplots show wide boxes, i.e., a more extensive data range, for the LGM period, characterized by fewer outliers, mostly from chronologies with sparse age control points and significant dating errors, than the periods with small box sizes.

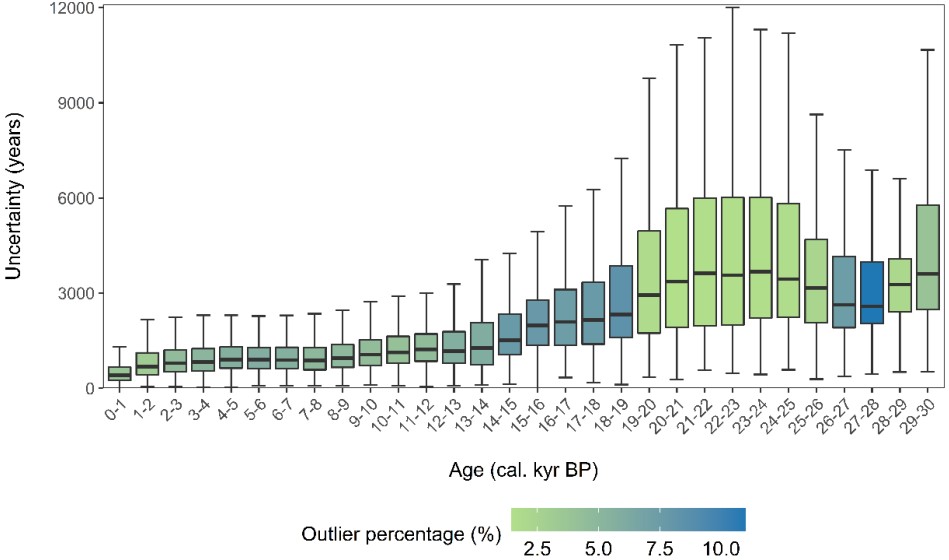

**Figure 6.** Boxplots of age uncertainties and outlier percentages in distinct time slices.

**3.3 Comparison of the LegacyAge 1.0 vs. original age-depth models**
For 906 records out of the 2831 records included in the LegacyAge 1.0, no calibrated chronologies were originally
available from the Neotoma and Cao et al. (2013, 2020) datasets for comparison. Of the remaining 1925 records,
the new LegacyAge 1.0 chronologies were selected instead of the original ones in 95.4% of cases, based on the
aforementioned criteria. However, some records still chose the original chronology, mainly because they are varve
chronologies, had incomplete metadata (e.g., missing sample depths), or included some non-[14]C dates that our
model could not accommodate (Supplement Table S6).
In most cases, the newly established chronologies were rather similar to the original ones. For 1012 records
(52.6% of 1925 records), the original chronologies were within the 95% confidence intervals of the LegacyAge
1.0 chronologies, while the other 913 records (47.4%) were partially or completely outside the 95% confidence
intervals.
Selected typical examples of the comparative results between the accepted LegacyAge 1.0 chronologies,
alternative newly generated but rejected chronologies, and the original chronologies are illustrated in Fig. 7. For
the *EL Tiro-Pass* record (ID 47502, Fig. 7a), both the original and LegacyAge 1.0 chronologies were established
by Bacon and are acceptable. However, the LegacyAge 1.0 chronology has the advantage that it makes use of the
latest radiocarbon calibration curve (IntCal20; Reimer et al., 2020), and the estimated surface age is more realistic
as sediments are still accumulating (Niemann and Behling, 2008). For the *Fargher Pond* record (ID 15344, Fig.
7b), the LegacyAge 1.0 chronology includes more varve ages from the Varved Sediments Database. These provide
a better constraint for the lowermost profile than the original model had (Grigg and Whitlock, 2002). For the
*Oltush Lake* record (ID 4320, Fig. 7c), the [14]C age of modern sediment in this lake is 350 yr BP and thus, the
assumption of a reservoir effect of 350 years resulted in slightly younger ages than originally given (Davydova
and Servant-Vildary, 1996). Some alternative rejected chronologies performed poorly due to the inability of high-
resolution Bacon models to accommodate accumulation rate changes (Fig.7b and Fig. 7c). Finally, for the
*Soppensee* record (ID 44723, Fig. 7d), most of the [14]C dates (> 540 cm) come from samples with insufficient
carbon to achieve accurate dating (Hajdas and Michczyński, 2010), and thus the original chronology, generated
from counting varves, outperformed our newly generated chronologies.

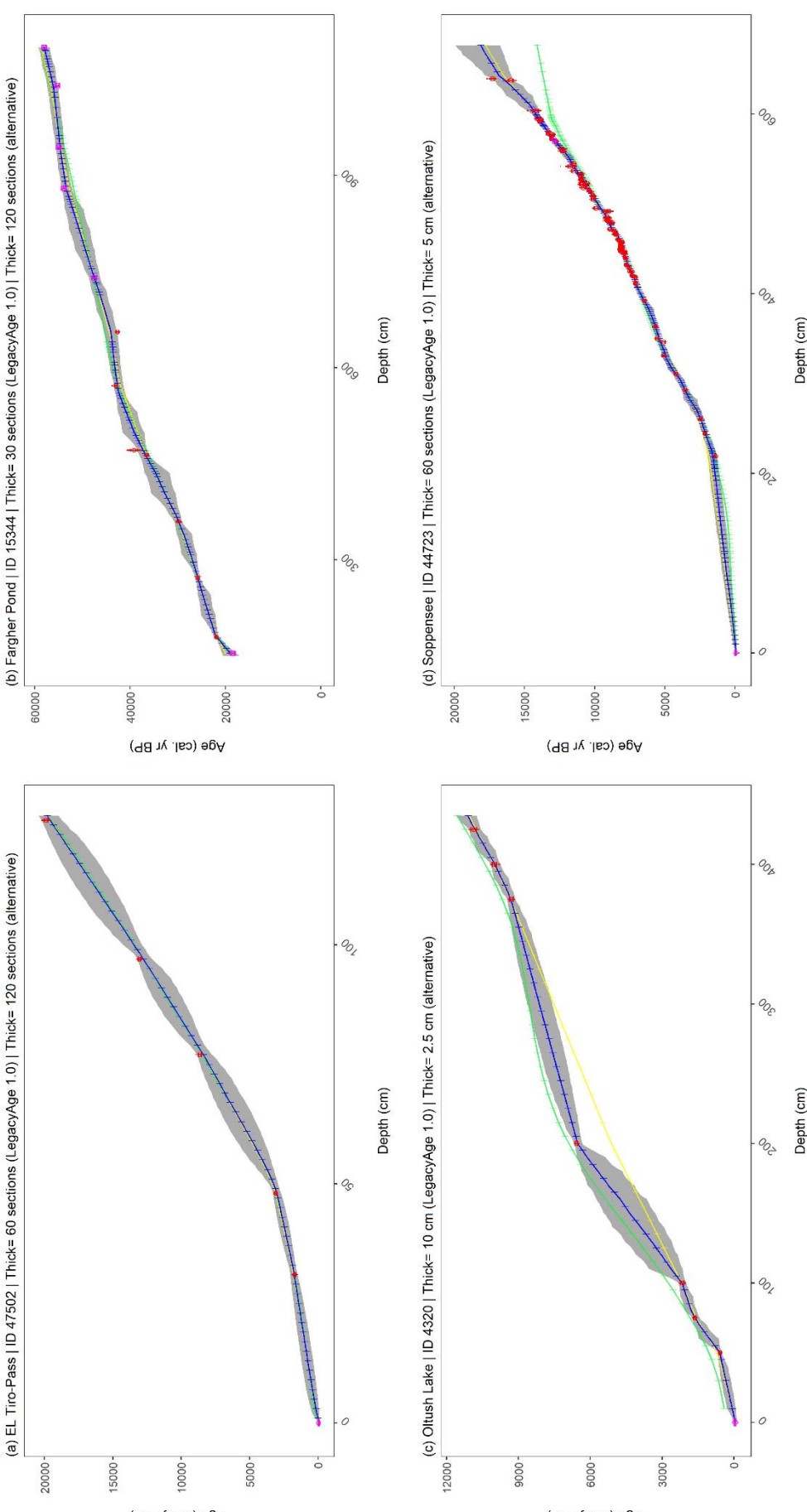

**Figure 7.** Comparison of LegacyAge 1.0 chronologies with the original ones. Green line: original chronology. Blue line: LegacyAge 1.0 chronology. Yellow line: alternative newly generated but rejected chronology. Red: date in chronology metadata. Pink: date from prior information. Grey shading: age uncertainties (95% confidence intervals).

**4 Code and data availability**

Seven supplementary datasets (Table S1-S7, in comma-separated values format) and one readme text about the LegacyAge 1.0 are accessible in the navigation bar 'Further details' of the PANGAEA page (https://doi.pangaea.de/10.1594/PANGAEA.933132; Li et al., 2021). We provided the chronological control points metadata (Table S1), prior information of dates from publication (Table S2), Bacon parameter settings (Table S3), original chronology metadata from the Neotoma and Cao et al. (2013, 2020) (Table S4), LegacyAge 1.0 chronology (Table S5), description of the comparison of original chronology and LegacyAge 1.0 (Table S6), and record references (Table S7) respectively. All datasets are already in long data format that can be joined by the dataset ID.

The R-code for calculation and comparison of chronologies with embedded manual, metadata for code runs, Bacon output graphs of each record, graphs comparison of original chronologies and LegacyAge 1.0, and a short shared-screen video of the R-code to show the usage on two example records are accessible on Zenodo (https://doi.org/10.5281/zenodo.5815192; Li et al., 2022).

**5 How to use the LegacyAge 1.0 dataset and code**

LegacyAge 1.0 provides the calibrated ages (mean, median, minimum, maximum) and uncertainties at each centimeter for each record with a 95% confidence interval (Supplement Table S5). All users can apply some interpolation algorithms in the chronologies, subsetted from the LegacyAge 1.0 dataset or outputted by our code, to assign ages for proxy depths of records.

As for the R-code, users only need to set the working directory where the Bacon results will be stored and input the record ID of interest to run it successfully. The manual and shared-screen video on R-code usage could provide helpful guidance for users, with or without some R-experience.

**6 Conclusion**

This paper presents the framework as well as metadata, machine-readable datings, R pipeline, chronologies, and age uncertainties of 2831 pollen records synthesized from the Neotoma Paleoecology Database (last access: April

2021) and 324 additional Asian records (Cao et al., 2013, 2020). Chronologies and uncertainties can be used for
synthesis works; metadata, datings, and pipelines can be used to reestablish the chronologies for customized
purposes, and the framework can be used to establish chronologies for newly updated records.

**Author contributions**. UH and CL designed the chronology dataset. CL and TB compiled the metadata and prior
information of the chronologies. AP and TB wrote the R scripts and ran the analyses under the supervision of UH
and CL. AMD contributed an initial R script for creating age-depth models with Bacon. CL wrote the first draft
of the manuscript under the supervision of UH. All authors discussed the results and contributed to the final
manuscript.
**Competing interests.** The authors declare that they have no conflict of interest.
**Acknowledgements.** The majority of data were obtained from the Neotoma Paleoecology Database
(http://www.neotomadb.org). The work of data contributors, data stewards, and the Neotoma community is
gratefully acknowledged. We would like to express our gratitude to all the palynologists and geologists who, either
directly or indirectly, contributed pollen data and chronologies to the dataset. We thank Andrej Andreev, Mareike
Wieczorek, and Birgit Heim from AWI for providing information on pollen records and data uploads. We also
thank Cathy Jenks for language editing on a previous version of the paper. This study was undertaken as part of
LandCover6k, a working group of Past Global Changes (PAGES), which in turn received support from the US
National Science Foundation, the Swiss National Science Foundation, the Swiss Academy of Sciences, and the
Chinese Academy of Sciences.
**Financial support.** This research has been supported by the European Research Council (ERC Glacial Legacy
772852 to UH), the PalMod Initiative (01LP1510C to UH), and the China Scholarship Council (201908130165
to CL).

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
