# Peer review of "Harmonized chronologies of a global late Quaternary pollen"

_Earth System Science Data, 2021_

## Author Comment (AC1)

**Response to comments of Anonymous Referee #1**

1. **General comments**

This paper describes a pollen records dataset, including explanations and descriptions of the dating methods involved in creating the dataset. The global coverage of this dataset is impressive and the presentation of the manuscript is quite good. There are some minor issues with accessing the data, and some considerable issues with the associated code attached to this paper.

*(1) While the general shape of the manuscript is good, I encourage a stronger focus on the data itself. These papers are most useful as upfront descriptions of data which requires a slightly different structure than a research articles. Specifically, I would recommend reshaping the intro and the abstract especially to put the data at the forefront, i.e. lead off with statements declaring the dataset, and what it is--for example, putting the name and description of the dataset as the first sentence in both.*

**Response: Thanks for your suggestion. We have changed the abstract section, i.e., 'We present a chronology framework named LegacyAge 1.0 containing harmonized chronologies for 2831 pollen records…', see line 15-17 of new text. But in the introduction section, we still follow the format of other articles in ESSD, i.e., describing the what, why, and how, to make the reading smoother.**

*(2) The description of dating methods needs to be expanded briefly, including explicitly defining terms such as "reservoir effect" or clarifying what "insufficient carbon" is. Lead dating is lacking description of methodology as is luminescence. Please also include how these dating methods add to measurement uncertainty in the data. Are uncertainties included?*

**Response: We have expanded the description of the dating methods, see section 2.2. The uptake of old carbon by aquatic plants or mosses or shells either originating from e.g., limestone in the catchment ('hard-water effect') or slow $^{14}$C exchange between the atmosphere and ocean interior can result in too old radiocarbon dates, which is called reservoir effect (see line 185-188). Conventional radiocarbon dating requires large amounts of carbon-containing material; otherwise, the dating may not be possible due to insufficient carbon. Furthermore, we have expanded the description of how these dating methods add to measurement uncertainty in the data. The uncertainty of date is available in supplement Table S1 at PANGAEA (https://doi.pangaea.de/10.1594/PANGAEA.933132).**

2. **Data (PANGAEA)**

*This dataset looks to be in good shape and is well-documented when i look at the site the DOI takes me too. When I download the .tab delimited file though, it is really tough to parse. Is there a reason this is in .tab format? A comma separated (.csv format) would be more universal, but I defer to the authors here if there is some subfield specific reason .tab format is better. Admittedly though, I found it difficult to work with this format when downloaded directly. The html web formatted table was easy enough to read.*

**Response: Seven supplementary datasets (Table S1-S7) and one readme text about the LegacyAge 1.0 are accessible in the navigation bar 'Further details' of the PANGAEA page (https://doi.pangaea.de/10.1594/PANGAEA.933132). As stated in the LegacyAge 1.0_readme.txt, all datasets were uploaded in .csv format, while the default format for PANGAEA storage is .tab. PANGAEA may rename the variables of the uploaded dataset to match its database format. However, these new variable names may have special characters that do not match the requirements of R, so we highly recommended downloading the original file we uploaded before running the R code.**

3. **Code**

*(1) The R code that accompanies this data paper and package is highly problematic from an open-code, data sharing perspective. It is formatted for personal use and not up to community standards. The main issue is the beginning call of `rm(list=ls())` This command cleans out and removes all entries in a user's memory and R workspace. Jenny Bryan wrote an excellent piece on why this snippet of code does not work for project based workflows (https://www.tidyverse.org/blog/2017/12/workflow-vs-script/)*

**Response: As you suggested we removed rm(list=ls()) memory clean. While coding, we were unaware that this could be a problem, so thanks for your input and the link to Jenny Bryans' work. Furthermore, we moved to store metadata, code, and results from GitHub to Zenodo (https://doi.org/10.5281/zenodo.5793936). Zenodo provides a persistent DOI to make the work easier to cite, supporting the data from Github repositories, as supported by referee 2.**

*(2) The major problem with this becomes apparent a couple of lines down when there are 'fixed' calls to data files that do not exist anywhere--nor can I find them. So running the code is impossible. I would recommend using URLs for those code calls so that when the code is run those data are imported directly from their fixed, online locations. The fixed DOIs from where your data are stored could be used.*

**Response: Thanks for your suggestion. We reduced the input files to three tables (Supplement Table S1, S3, and S4) defined in the first 51 rows of code together with an embedded manual. We used URLs for those code calls so that when the code is run those three input files are imported directly from PANGAEA (https://doi.pangaea.de/10.1594/PANGAEA.933132). Also, all readers can download these files from PANGAEA or Zenodo (https://doi.org/10.5281/zenodo.5793936) to a new folder and insert the path of the folder to the folder definition at the begin of the code.**

*(3) This area of this manuscript/data must be addressed. Additionally, the code is commented adequately, and follows a fairly good syntax, formatting structure. I applaud that. The repo in GitHub though has no readme and no documentation there. That really needs to be added. You could include a lot of what is in this paper, in the data metadata write up elsewhere. I would also encourage including a copy of this manuscript as well as copious amounts of links.*

*A big ask, which I think would take this next level, is to include a vignette or markdown file showing how to work with his data that includes a small, worked example.*

*In the current state, I cannot run the code, which gives me pause on my recommendation.*

**Response: We apologize again for this. Additional to the embedded manual, we provided a short shared-screen video in Zenodo (https://doi.org/10.5281/zenodo.5793936) to show the usage on two example sites. The embedded manual and the screen video should be helpful as readme/documentation, and now it should be possible to run the code easily.**

4. **Specific comments**

   *(1) line 44 - the phrase "calibrated and uncalibrated" is confusing.*

   **Response: Calibrations of radiocarbon age determinations are applied to convert the conventional radiocarbon age to calendar years. Thus, uncalibrated $^{14}$C age is the conventional radiocarbon age, calibrated $^{14}$C age is the calendar age. We deleted this sentence.**

   *(2) line 65-75 - it would be advisable to have these variables in a table with further descriptions.*

   **Response: Seven supplementary datasets (Table S1-S7) and one readme text about the LegacyAge 1.0 are available at PANGAEA (https://doi.pangaea.de/10.1594/PANGAEA.933132). Supplement Tables S1 and S4 include all metadata of chronological control points and original chronology from Neotoma and Cao et al. (2013, 2020). Readers can read the readme text to understand the variables better.**

*(3) line 79-80 - repeated use of references to "most common"*

**Response: Sorry, this comment is confusing to understand; refer to the repeated use of the word "most common" or the reference? We deleted this word in the new manuscript. But for this reference (Roberts, 2013), various dating methods have been summarized. In expanding the description of the dating methods, we have also added some additional references (see line 150).**

*(4) Section 2.3.1. - for this type of paper, consider leading this section off with what you have as your final sentence, then describing it. "...all age relationships in our data set are constructed using Bacon..." then describe why and what and how.*

**Response: As you suggested, we led this section with the final sentence: 'We used the Bacon framework as it is one of the most commonly used methods for age-depth modeling...', see line 120-135.**

*(5) line 139-141 - where did the latest calibration curves come from? this sentence lacks context.*

**Response: The latest calibration curves, approved by the radiocarbon community, are stored in Bacon. Readers also can visit them at http://calib.org/. The latest calibration curves (IntCal20, SHcal20, and Marine20) were released in 2020, see line 149-150.**

*(6) Section 2.3.4 consider laying this section out using bullets or with some kind of work design flow infographic.*

**Response: We laid this section out using bullets following your suggestion, see line 163-181.**

*(7) \* just a note format your units with super- and subscripts, not / notation*

**Response: We changed '/' to superscript ($^{-1}$).**

*(8) lines 167 -Consider again bullets or something instead of a numbered list inside of a paragraph.*

**Response: We laid this section out using bullets following your suggestion, same as before, see line 185-206.**

---

## Author Comment (AC2)

**Response to comments of Anonymous Referee #2**

**1. General comments**

Most analyses using Neotoma or other archived pollen data are dependent, at least to some extent, on the chronologies. The available chronologies have variable quality: some record have an uncalibrated chronology, others have a Bayesian chronology. In many cases the uncertainty on the chronology is not available, or if it is, just the upper and lower credibility interval. To synthesise pollen data from several datasets, it may be necessary to harmonised the age-depth models, a huge amount of work. Once such harmonisation is presented in this current manuscript.

*(1) As far as I can tell, the chronologies are not archived, but instead the metadata needed to make the chronologies. This is probably a good idea as it encourages the user to check the parameters.*

**Response: Seven supplementary datasets (Table S1-S7, in comma-separated values format) and one readme text about the LegacyAge 1.0 are accessible in the navigation bar 'Further details' of the PANGAEA page (https://doi.pangaea.de/10.1594/PANGAEA.933132). We provided the chronological control points metadata (Table S1), prior information of dates from literature (Table S2), Bacon parameter settings (Table S3), original chronology metadata from the Neotoma and Cao et al. (2013, 2020) (Table S4), LegacyAge 1.0 chronology (Table S5), description of the comparison of original chronology and LegacyAge 1.0 (Table S6), and record references (Table S7) respectively. Furthermore, the R-code for calculation and comparison chronologies with embedded manual, metadata for code runs, Bacon output graphs of each record, graphs comparison of original chronologies and LegacyAge 1.0, and a short shared-screen video of the R-code to show the usage on two example records are accessible on Zenodo (https://doi.org/10.5281/zenodo.5793936). Thus, readers can obtain chronologies directly using Table S5 we provided or use the script to calculate several or all of the records they are interested in. We also encourage readers to check the parameter settings.**

*(2) One important result is that "95.4% of records could be improved ". However, it is unclear what objective criteria were used to determine whether the new model represented an improvement. The criteria need to be explicitly stated.*

**Response: The criteria for the preferred models are that the model fitted the dates well, had small uncertainties, combined dates with prior information (e.g., geological and hydrological setting, environmental history), and calibrated with the latest calibration curves, see line 218-225 of new text. Our newly harmonized chronological framework uses the latest calibration curves and Bayesian statistics and fully considers the prior information collected from the original literature.**

*(3) The metadata and code are available on github (Zenodo.org would be preferable for the final version).*

**Response: We agree with your suggestion. As we know, Github is the largest and most common source code host. Zenodo provides a persistent DOI to make the work easier to cite, supporting the data from Github repositories. Therefore, we moved to store metadata, code, and results from GitHub to Zenodo (https://doi.org/10.5281/zenodo.5793936).**

2. **Data (PANGAEA)**

*(1) The data are arranged in wide format, with a set of columns for each date. This is not the ideal way to arrange the data, as it makes the code much more complicated to deal with this structure, and will need extra extra columns adding in the future to cope with new sites. A better setup would be to have the data in long format, perhaps in multiple files that can be joined by the dataset ID.*

**Response: All datasets are already in long data format, i.e., there are fewer variables than observations. For example, there are 23 variables and 19,990 observations in supplement Table S1 at PANGAEA (https://doi.pangaea.de/10.1594/PANGAEA.933132). While wide data formats take less memory and are easier to analyze, long data formats make adding or subtracting observations easier, as you agreed. If we want to use the ggplot2, an open-source data visualization package for the statistical programming language R, for plotting, the long data format is the best choice.**

*(2) At present, datasets are marked as being marine or otherwise. At least in principle, there could be datasets where some dates are on marine fossils, and others on terrestrial macrofossils. Marine should be a property of the date, not the core.*

**Response: Yes, you are right. The sites marked as marine or otherwise just distinguish the calibration curve used. Radiocarbon dates of a terrestrial and marine organism of equivalent age have a difference of about 400 radiocarbon years. Terrestrial and marine samples cannot be compared or associated without accounting for the marine radiocarbon reservoir effect. Therefore, we used a different calibration curve for dates on marine fossils than terrestrial. In addition, after rechecking the sites marked as marine in the dataset, all the dating materials are from marine sediments.**

*(3) Setting a constant reservoir age for a core without error seems risky to me. Do you check these are only occuring with bulk dates, as there should not normally be a problem with terrestrial macrofossils.*

**Response: Although the reservoir ages with errors would be better, the values reported by the original author through various methods are mostly constant. We fully respect the original authors' comments because we are no more familiar with the sites than they are. As you can read from the variable 'Material_Dated' in supplement Table S1, there are a variety of dating materials. Aquatic plants, shells, and bulk sediments that may be affected by reservoir effects were often used for radiocarbon dating if ideal dating materials such as terrestrial plant macrofossils have insufficient carbon content for dating. Although the accelerator mass spectrometry (AMS) technique requires much smaller quantities of carbon than the conventional radiocarbon dating, a tiny amount of contamination is sufficient to affect a date. Thus, some dates from terrestrial plant macrofossils mixed with other materials also report radiocarbon reservoir effects.**

**3. Code**

*(1) The code is presented in a single script. This is fine if someone wants to make chronologies for all datasets, but often people will want to run a subset of the sites, and may find the script difficult to use. Breaking the script into functions than each do one task, perhaps compiled into a small package with help files and other documentation, would make the product more useful.*

**Response: Thanks for your suggestion. We reorganized the code and reduced the input files to three tables (Supplement Table S1, S3, and S4) defined in the first 51 rows of code together with an embedded manual. The reader can calculate chronology for all records or some records of interest by modifying lines 35-36. We used URLs for those code calls so that when the code is run, those three input files are imported directly from PANGAEA (https://doi.pangaea.de/10.1594/PANGAEA.933132). Also, all readers can download these files from PANGAEA or Zenodo (https://doi.org/10.5281/zenodo.5793936) to a new folder and insert the path of the folder to the folder definition at the begin of the code. Additional to the embedded manual, we provided a short shared-screen video in Zenodo to show the usage on two example sites. The embedded manual and the screen video should be helpful as readme/documentation, and now it should be possible to run the code easily. Splitting the code into separate functions is an interesting idea, but getting this to universal usage (now it is very specific and focused on our data) would need more modifications. But to consider this, we separated the "Age Allocation" part into a standalone code that should be the most valuable part of the main script for other purposes.**

*(2) The code includes fixes for issues the authors found in Neotoma. Please report these issues, and make sure the code to fix them (e.g. line 187) will work safely when the error is fixed.*

**Response: We excluded the Neotoma downloads from the code because all information needed is located in three tables (Supplement Table S1, S3, and S4), and for them, we can be sure about the data. It would not be possible to reproduce our results with changes in the Neotoma data or the Neotoma package. An automatic modeling with data from Neotoma would be very nice, but without more input information (which we collected in the parameter table) and only standard parameters, the results would not be very satisfying. Actually, the fixes in the code had been implemented just in the case of unit mistakes, and for one site, we contacted the Neotoma team.**

**4. Figure**

*(1) Figures 4-6. The x-axis, labelled 0-1, 1-2 etc, is rather cluttered and difficult to read. Please consider other ways to label them. The relative lack of outliers in the pre-LGM sediment is interesting, but probably represents over-fitting the models to sparse data.*

**Response: The label of the X-axis is changed to not fully displayed, see revised figures 4-5. Skewed distribution of data usually leads to more outliers in the boxplot. As you can read from the following figure (Fig. R1), a time slice includes more outliers if the data are negatively skewed distribution.**

[Figure]

**Figure R1. Boxplots of age uncertainties and outlier percentages in distinct time slices (with outliers)**

*(2) Figure 7 has a minuscule font: I need to view it at 200% before I can read it. No need to repeat the legend four time - once is enough and leaves more space for everything else.*

**Response: As suggested by reviewer #3, we have newly provided the section thicknesses in the title of each figure, 'best' and 'poor' separately, see revised figure 6. Also, one additional age-depth realization with relatively poor performance has been included for comparison. Preference was given to models that fitted the dates well, had small mean uncertainties, and good runs of Markov Chain Monte Carlo iterations (i.e., a stationary distribution with little structure among neighboring iterations as indicated by the traceplot of the joint likelihood) when choosing the 'best' model for each record.**

**5. Specific comments**

*Line 202 "with fewer than 2" Maybe rewrite as "only one" for clarity. Consideration should be given as to the minimal number of dates that can give a good chronology - I would be cautious using a model based on only two dates.*

**Response: It has been evaluated that some records do not have one reliable date, and some may have only one, so 'with fewer than 2' may be more appropriate. As we know, the more date a chronology has, the more reliable it is. As you read from column *'Dates_Number'* of the supplement Table S5, 9.7% of the records have only two dates. We have kept these to keep as many records as possible.**

---

## Author Comment (AC3)

**1. General comments**

In this work the authors present their attempt to harmonize mainly radiocarbon-based chronologies of continental climate records. The harmonisation is with respect of age-model software usage, calibration curve usage, which is a very valuable task. Furthermore, harmonisation is performed with respect to parameters used for the age-depth modelling software. As far as I understand, the authors use the age-modelling software Bacon for age-depth modelling of a huge quantity of records. Before modelling, the cores were manually evaluated in terms of complications, such as radiocarbon reservoir effects, water lines, etc.

While I appreciate their approach, I think there are some things to be improved before suggesting this piece of work for publication.

**2. Data (PANGAEA)**

*(1) Furthermore, I am not able to find age-depth profiles on their provided Pangaea-page. I thought the authors did all their work (handling reservoir effects, water lines, deciding for the best thicknesses to be applied, …) in order to provide a homogeneous age-depth data set. And according to their paper, they spend a lot of efforts to evaluate the datings etc of all records. It would be a pity, if they would not share this. Or is the user supposed to start from scratch again? Even if it 'only' means to run their script – if I understand the code structure correctly, the user has to run all of their thousands of records, even if the user is only interested in one or two records. Especially, as this means to run 'millions of MCMC iterations' (line 120) which cannot be that cheap as even admitted by the authors: "… it needs much supervision and computing power" (line 122). Why not provide all age depth models (including uncertainties) in addition to all meta data and code? Or at least enable the user to only calculate the age-depth models of the records they are interested in?*

**Response: Seven supplementary datasets (Table S1-S7, in comma-separated values format) and one readme text about the LegacyAge 1.0 are accessible in the navigation bar 'Further details' of the PANGAEA page (https://doi.pangaea.de/10.1594/PANGAEA.933132). We provided the chronological control points metadata (Table S1), prior information of dates from literature (Table S2), Bacon parameter settings (Table S3), original chronology metadata from the Neotoma and Cao et al. (2013, 2020) (Table S4), LegacyAge 1.0 chronology (Table S5), description of the comparison of original chronology and LegacyAge 1.0 (Table S6), and record references (Table S7) respectively. Furthermore, the R-code for calculation and comparison chronologies with**

embedded manual, metadata for code runs, Bacon output graphs of each record, graphs comparison of original chronologies and LegacyAge 1.0, and a short shared-screen video of the R-code to show the usage on two example records are accessible on Zenodo (https://doi.org/10.5281/zenodo.5793936). Note that we moved to store them from GitHub to Zenodo. Zenodo provides a persistent DOI to make the work easier to cite, supporting the data from Github repositories, as supported by referee 2. Thus, readers can obtain chronologies directly using Table S5 we provided or use the script to calculate several or all of the records they are interested in. We also encourage readers to check the parameter settings.

*(2) Another critical question is about the final age models. As I cannot find them, nor are able to run the R script, I have to ask: Which depths intervals do you choose to save for the homogenised age-depth models? In the paper you mention the effect of choosing different levels or depth intervals on the goodness of the model data and that some are better suited than others. However, I even wonder, why a user should care about having the age-depth relationship on a fixed sampling interval? If I want to work with other paleoclimate data, I am interested in an age-depth model, which provides dates at depth, where the proxies were measured. Is the output of your script arranged in a way, that this could be easily accessed? Unfortunately, this is not mentioned in the paper. Or do you expect the user to apply some (more or less) fancy interpolation algorithm to assign ages for the proxy depths?*

Response: Two Bacon parameters need to be clarified, 'thick' and 'd.by'. Bacon will divide the core into many vertical sections (by default of thick=5 cm thickness) during calculation, which significantly affects the flexibility of the age-depth model. Since our dataset contained 2831 records, it was unrealistic to establish the age-depth relationship for each record using different section thicknesses. To batch process, we finally selected six thicknesses tested many times for most chronologies (ca. 85%). We have also adjusted the section thickness for some records, please refer to supplement Table S3. Of course, the reader can check or modify the parameter settings to generate a higher quality chronology. Another parameter, 'd.by,' i.e., depth intervals at which ages are calculated. Table S5 includes all records' the calibrated ages (mean, median, minimum, maximum) at each centimeter. Readers can assign ages for the proxy depths in two ways: applying the interpolation algorithm in the results we provided, or modifying the parameter 'd.by' (default=1 cm) to recalculate it. In summary, the six section thicknesses (2.5 cm, 5 cm, 10 cm, 30 sections, 60 sections, and 120 sections) mentioned in the manuscript affect the flexibility of the age-depth model, which is different from the depth interval of the chronology.

**3. Code**

*Usually, such a data set and code is generated to be used. Unfortunately, I cannot find any description or manual, how to access the age-depth models. Nor is it possible for me to run the R-script. I admit, I am a R-noob, but I think, application should be properly described with at least a short manual for users with some R-experience (or even noobs). This does not have to come with this publication, but it should at least appear on their github space next to the R-file.*

**Response: We apologize again for this. We reorganized the code and reduced the input files to three tables (Supplement Table S1, S3, and S4) defined in the first 51 rows of code together with an embedded manual. The reader can calculate chronology for all records or some records of interest by modifying lines 35-36. We used URLs for those code calls so that when the code is run, those three input files are imported directly from PANGAEA (https://doi.pangaea.de/10.1594/PANGAEA.933132). Also, all readers can download these files from PANGAEA or Zenodo (https://doi.org/10.5281/zenodo.5793936) to a new folder and insert the path of the folder to the folder definition at the begin of the code. Additional to the embedded manual, we provided a short shared-screen video in Zenodo to show the usage on two example sites. The embedded manual and the screen video should be helpful as readme/documentation, and now it should be possible to run the code easily.**

**4. Figure**

*Fig. 7: Please provide information about which of the twelve generated age-depth models for each record you show here! Would it be possible to show one additional age-depth realisation, which fits less good with the measured ages. Only to give the reader an idea about the effects of the choice of depths intervals.*

**Response: Thanks for your suggestion. As you read from revised Figure 7, there are two section thicknesses in the title of each figure, 'best' and 'poor' separately. Also, one additional age-depth realization with relatively poor performance has been included for comparison. Preference was given to models that fitted the dates well, had small mean uncertainties, and good runs of Markov Chain Monte Carlo iterations (i.e., a stationary distribution with little structure among neighboring iterations as indicated by the traceplot of the joint likelihood) when choosing the 'best' model for each record.**

**5. Specific comments**

*(1) L16 and 46: Please elaborate a bit more on what you understand by 'harmonized chronology' already this early in the manuscript. I am pretty, sure, that different people understand different things under this term. I mean later in the paper it becomes clear, what you understand by this term, but I think it is worth to highlight this already in the beginning of your work.*

**Response: As you understand, 'harmonized chronology,' i.e., using the same strategy for consistent inference of age and age uncertainty. We also elaborated this term a bit more in the introduction section, see line 48 of new text.**

*(2) L27-28: This sentence needs more explanations. Maybe not here in the abstract, but below in the according text passages. Please find a more detailed comment below.*

**Response: Yes, only the final result of the comparison is only shown here. The criteria for the preferred models are that the model fitted the dates well, had small uncertainties, combined dates with prior information (e.g., geological and hydrological setting, environmental history), and calibrated with the latest calibration curves, see line 222-225.**

*(3) L69-74: You provide quite some detailed information on metadata, which I appreciate a lot. However, I doubt that putting those data all in one file is the best option. I agree with referee 2 to splitting this file up in several is maybe more appropriate and easier to handle. At least keep this in mind for any potential future improvements.*

**Response: We reorganized the metadata into three supplement tables: the chronological control points metadata (Table S1), Bacon parameter settings (Table S3), and the original chronology metadata from the Neotoma and Cao et al. (2013, 2020) (Table S4). Also, readers can learn more information about the variables in the table from the readme text.**

*(4) L155: 'acc.mean' is possibly 'acc.rate'?*

**Response: The correct abbreviation for mean accumulation rate is 'acc.mean', see line 164. We have made the change in the text and apologize for the confusion.**

*(5) L158: 'We tested six thicknesses (2.5 cm, 5 cm, 10 cm, 30 sections, 60 sections, and 120 sections) …'. I am not very familiar with Bacon. But, why would you want to test those 6 sampling intervals? I mean, the proxies of the cores were measured at specific depths - wouldn't it be more suitable to only interpolate to those depths, where proxy data exist? Actually, this is the data, I would be interested in. But it seems, that this is missing completely. What do you suggest to finally obtain the ages at those depths?*

**Response: As mentioned earlier, these six section thicknesses (2.5 cm, 5 cm, 10 cm, 30 sections, 60 sections, and 120 sections) affect the flexibility of the age-depth model, which is different from the depth interval of the chronology. All readers can assign ages for the proxy depths in two ways: applying the interpolation algorithm in the results we provided, or modifying the parameter 'd.by' to recalculate it. For example, we assigned ages to pollen samples by interpolation (https://doi.pangaea.de/10.1594/PANGAEA.929773). As you can read from the supplement Table S5, we provided the estimated age at each centimeter, which provides the possibility for other proxies (not only pollen) to interpolate ages at a specific depth.**

*(6) L159: 'artificial surface age', Why would it be necessary to add an artificial date? I don't know if I understand the concept of adding an artificial date correctly. Stating things like this sounds very arbitrary. Or do you mean you added another age-constraint due to the assumption that the core sedimentation was active until core recovery? And that the additional age constraint is the year of core recovery? If yes, please consider to specify accordingly.*

**Response: Yes, you are right. If the core was collected from sites where sediment was still accumulating, the core-top age could be one significant time control for the chronologies. Therefore, an estimated artificial surface age (-50 + -30 cal yr BP) was used if the core-top age cannot be obtained from the sampling date in literature or original chronology in Neotoma. We have also made changes in the text, see line 137-147.**

*(7) L159: 'generating 12 age models for each core'. Just to make sure I understand correctly. Your code provides 12 age-depth models for one core. Are all provided in output files?*

**Response: Yes, our code initially outputs 12 age-depth models for each record. We only provided the 'best' chronology for each record to PANGAEA (https://doi.pangaea.de/10.1594/PANGAEA.933132; Supplement Table S3), also the Bacon output graphs of each record in Zenodo (https://doi.org/10.5281/zenodo.5793936). You will get the best chronology for each record if you run the script directly. Meanwhile, if you want to get multiple age-depth models for each record, you can do so by modifying the column *'Resolution.cm'* or *'Resolution.section'* of Table S3. The embedded manual and the screen video should be helpful as readme/documentation, and now it should be possible to run the code easily.**

*(8) L170: I think, C exchange between dissolved C-species in water and atmospheric CO2 is not responsible for 'too old radiocarbon dates'. Instead, this process counter balances to some degree the effect of the arguments listed earlier in this sentence.*

**Response: We agree with you. However, slow $^{14}$C exchange between the atmosphere and ocean interior, can result in too old radiocarbon dates, which we originally wanted to express.**

**Radiocarbon dates of a terrestrial and marine organism of equivalent age have a difference of about 400 radiocarbon years, i.e., marine radiocarbon reservoir effect, see line 185-188.**

*(9) L171-173: For some records you added your evaluation of reservoir effects. I appreciate this a lot, but I think it is worth to add a column in your metadata file and mark those records. This would allow a better transparency about what is your evaluation and which information came from the original studies.*

**Response: Readers can view this information in column *'Reservoir'* of supplement table S3, or view type 2 in column *'Category'* of supplement table S2 to learn how this information was obtained.**

*(10) L184: For the use of radiocarbon dates for modelling purposes, you followed 'in most cases the suggestions in the original publications'. Please consider – again for a better transparency - to provide information (maybe in your metadata file), for which records you did not follow the suggestions of the original publications.*

**Response: We rejected or added dates based on prior information collected from the original publications and Neotoma. As you can read from the last two columns of supplement Table S2, all kinds of prior information are listed here.**

*(11) L189-191: 'For each record, 12 age models were visually assessed. Preference was given to models that fitted the dates well and with small uncertainties when choosing the 'best' model for each record (Blaauw and Christen, 2011; Blaauw et al., 2018).'. This is a lot of work for thousands of records. You are sure, that you did this all correctly for this large amount of records? I wonder if it would have been more objective to apply a short statistical test on this. I mean, most likely a simple least square test between age model and ages of dated depths would do a better and faster job. Also the 'small uncertainty' argument would be most likely more precise and faster to obtain, when calculating the mean uncertainty instead relying on visual assessment.*

**Response: Yes, you are right. We visually evaluated the 12 initial age-depth models for each record following the Bacon manual's common method, which took a lot of time. Preference was given to models that fitted the dates well, had small mean uncertainties, and good runs of Markov Chain Monte Carlo iterations (i.e., a stationary distribution with little structure among neighboring iterations as indicated by the traceplot of the joint likelihood) when choosing the 'best' model for each record. This work was visually evaluated by two individuals independently according to a unified standard, and then the results of both were combined to reduce the error. We also tried least-squares initially, but it is dangerous to choose the best model only based on its results. A significant disadvantage of the least-squares method is that it is greatly affected by the**

disturbance of outliers. If we only choose a model with the least-squares, this model may have significant uncertainties due to the overfitting dates. Finally, we decided to use the visual method because Bacon's output of the graph is apparent. Specifically, we can quickly check the result of Markov Chain Monte Carlo iterations, the overall picture of uncertainty, and how well the model fits the date. If the model can fit the date well, it is actually also an application of the least-squares idea, but we judge subjectively rather than statistically. We also calculated the mean uncertainty of each model for each record. The reader can view the mean uncertainty (95% confidence ranges) of the 'best' model and uncertainty at each centimeter of each record at supplement Table S5.

*(12) L203: Who did the evaluation about what a reliable date is? You or the original authors? I can imagine, that this is a difficult task, especially for cores from others.*

**Response: We assessed all dates based on prior information, as authors usually report all $^{14}$C dates from a sequence, even if some are deemed inaccurate. We also fully respect the original authors' comments because we are no more familiar with the sites than they are.**

*(13) L247-248: 'where original chronologies outperformed LegacyAge 1.0, ...' How do you know, which model approach outperforms the other? How can you measure or evaluate this? Do you have knowledge of the 'true sedimentation history' of all those records to be able to judge this? Which one do you choose from your 12 ones/core? I think it is very crucial to provide more details on this issue. Or, in case you wanted to express a different thing with this expression, please consider to rephrase this sentence.*

**Response: The newly generated 'best' calibrated chronology of each record were compared with original chronologies taken from the Neotoma and Cao et al. (2013, 2020) datasets (Supplement Table S4) to evaluate the performance of the new models. The criteria for the preferred models are that the model fitted the dates well, had small uncertainties, combined dates with prior information (e.g., geological and hydrological setting, environmental history), and calibrated with the latest calibration curves. We have added it to section 2.4 in the text (line 222-225).**

---

## Author Comment (AC4)

**Response to comments of Anonymous Referee #1**

**1. General comments**

Reviewer comment: This paper describes a pollen records dataset, including explanations and descriptions of the dating methods involved in creating the dataset. The global coverage of this dataset is impressive and the presentation of the manuscript is quite good. There are some minor issues with accessing the data, and some considerable issues with the associated code attached to this paper.

Reviewer comment: *(1) While the general shape of the manuscript is good, I encourage a stronger focus on the data itself. These papers are most useful as upfront descriptions of data which requires a slightly different structure than a research articles. Specifically, I would recommend reshaping the intro and the abstract especially to put the data at the forefront, i.e. lead off with statements declaring the dataset, and what it is--for example, putting the name and description of the dataset as the first sentence in both.*

**Response: Thank you for your suggestion. We have revised the abstract section according to your suggestion.**

**New text (line 15-17): 'We present a chronology framework named LegacyAge 1.0 containing harmonized chronologies for 2831 pollen records (downloaded from the Neotoma Paleoecology Database and the supplementary Asian datasets) together with their age control points and metadata in machine-readable data formats'.**

**The introduction is now in line with other ESSD papers on similar topics (Sánchez Goñi et al., 2017; Cao et al., 2020), i.e., 1. introducing the potential of available pollen databases, 2. stating the research gap why pollen databases cannot be fully exploited (i.e., missing harmonized chronologies), 3. summarising this study.**

Reviewer comment: *(2) The description of dating methods needs to be expanded briefly, including explicitly defining terms such as "reservoir effect" or clarifying what "insufficient carbon" is. Lead dating is lacking description of methodology as is luminescence. Please also include how these dating methods add to measurement uncertainty in the data. Are uncertainties included?*

**Response: We have expanded the description of the dating methods, and how these dating methods add to measurement uncertainty in the data, please see section 2.2.**

**New text (line 187-190): 'Reservoir effects: the uptake of old carbon by aquatic plants, mosses, or shells either originating from, e.g., limestone in the catchment ('hard-water effect') or slow $^{14}C$**

**exchange between the atmosphere and ocean interior, can result in too old radiocarbon dates (Philippsen, 2013; Philippsen and Heinemeier, 2013; Giesecke et al., 2014; Heaton et al., 2020)'.**

**New text (line 84-96):**

**'*Radiocarbon dating:* most records were dated using radiocarbon-based methods ([14]C dating, conventional or accelerator mass spectrometry, Christie, 2018), covering the time range of ca. the last 50 kyr BP (before present, where 'present' is 1950 CE). However, the accuracy and precision of the radiocarbon dates depend on the calibration curve, taphonomy, and dating materials (Blois et al., 2011; Heaton et al., 2021).**

***Lead-210 dating:* the uppermost part of some lake records has been dated using a radioactive isotope of lead (lead-210), which has a half-life of ca. 22 years and provides useful age control for the last 75-150 years. However, the abundance of other radioactive isotopes (e.g., Caesium-137) affects the accuracy and precision of the calibration curve for lead-210, resulting in temporal uncertainty (Appleby and Oldfield, 1978; Cuney, 2021).**

***Luminescence dating:* archaeological materials, loess, and river sediments have often been dated via luminescence, including thermoluminescence (TL) and optically stimulated luminescence (OSL), which cover time scales from millennia to hundreds of thousands of years (Roberts, 2013). Due to the systematic and random errors in the measurement process, the luminescence ages have at least 4-5% uncertainty, which widens with increasing time (Wallinga and Cunningham, 2015)'.**

**2. Data (PANGAEA)**

Reviewer comment: *This dataset looks to be in good shape and is well-documented when I look at the site the DOI takes me too. When I download the .tab delimited file though, it is really tough to parse. Is there a reason this is in .tab format? A comma separated (.csv format) would be more universal, but I defer to the authors here if there is some subfield specific reason .tab format is better. Admittedly though, I found it difficult to work with this format when downloaded directly. The html web formatted table was easy enough to read.*

**Response: Thanks for your suggestion. All datasets can be downloaded in .csv and .tab format. We improved the documentation on how to access the dataset in 'Code and data availability' and in the readme text.**

**New text (line 313-320): 'Seven supplementary datasets (Table S1-S7, in comma-separated values format) and one readme text about the LegacyAge 1.0 are accessible in the navigation bar**

**‘Further details’ of the PANGAEA page (https://doi.pangaea.de/10.1594/PANGAEA.933132; Li et al., 2021a). We provided the chronological control points metadata (Table S1), prior information of dates from publication (Table S2), Bacon parameter settings (Table S3), original chronology metadata from the Neotoma and Cao et al. (2013, 2020) (Table S4), LegacyAge 1.0 chronology (Table S5), description of the comparison of original chronology and LegacyAge 1.0 (Table S6), and record references (Table S7) respectively. All datasets are already in long data format that can be joined by the dataset ID’.**

**Readme text: ‘Please select 'Further details' on the left navigation bar of the webpage to access the dataset in .csv format’. The 'Download Data' on the bottom navigation bar of the page can also download the dataset as tab-delimited text and view the dataset as HTML (shows only the first 2000 rows), stored in the PANGAEA. However, PANGAEA may rename the variables of the uploaded dataset to match its database format. These new variable names may have special characters that do not match the requirements of R, so we highly recommended downloading the original file (in .csv) we uploaded before running the R code’.**

**3. Code**

Reviewer comment: *(1) The R code that accompanies this data paper and package is highly problematic from an open-code, data sharing perspective. It is formatted for personal use and not up to community standards. The main issue is the beginning call of `rm(list=ls())` This command cleans out and removes all entries in a user's memory and R workspace. Jenny Bryan wrote an excellent piece on why this snippet of code does not work for project based workflows (https://www.tidyverse.org/blog/2017/12/workflow-vs-script/)*

**Response: We revised the code according to your suggestion, i.e., we removed rm(list=ls()) memory clean. While coding, we were unaware that this could be a problem, so thanks for your input and the link to Jenny Bryans’ work. Furthermore, we now store metadata, code, and results on Zenodo (https://doi.org/10.5281/zenodo.5815192), which better supports open-coded and data sharing.**

Reviewer comment: *(2) The major problem with this becomes apparent a couple of lines down when there are 'fixed' calls to data files that do not exist anywhere--nor can I find them. So running the code is impossible. I would recommend using URLs for those code calls so that when the code is run those data are imported directly from their fixed, online locations. The fixed DOIs from where your data are stored could be used.*

**Response: Thanks for your suggestion. We revised the code and reduced the input files to three tables, defined in the first 51 rows of code. We also used URLs for the calls to import these three input files directly from PANGAEA (Supplement Table S1, S3, and S4; https://doi.pangaea.de/10.1594/PANGAEA.933132).**

**New code (line 38-51):**

'metadata <- read.csv2("https://download.pangaea.de/reference/111158/attachments/Table-S1_chronological_control_points_metadata.csv", stringsAsFactors = FALSE, sep = "\t", dec = ".")

parameter <- read.csv2("https://download.pangaea.de/reference/111160/attachments/Table-S3_bacon_parameter_settings.csv", stringsAsFactors = FALSE, sep = "\t", dec = ".")

AgeDepthPollen <- read.csv2("https://download.pangaea.de/reference/111161/attachments/Table-S4_original_chronology_metadata_by_pollen_records.csv",stringsAsFactors = FALSE, sep = "\t", dec = ".")'

Reviewer comment: *(3) This area of this manuscript/data must be addressed. Additionally, the code is commented adequately, and follows a fairly good syntax, formatting structure. I applaud that. The repo in GitHub though has no readme and no documentation there. That really needs to be added. You could include a lot of what is in this paper, in the data metadata write up elsewhere. I would also encourage including a copy of this manuscript as well as copious amounts of links.*

*A big ask, which I think would take this next level, is to include a vignette or markdown file showing how to work with his data that includes a small, worked example.*

*In the current state, I cannot run the code, which gives me pause on my recommendation.*

**Response: We apologize for this. We revised the code and rephrased this part of the text.**

**Description of the uploaded file in new text (line 321-324):**

'The R-code for calculation and comparison of chronologies with embedded manual, metadata for code runs, Bacon output graphs of each record, graphs comparison of original chronologies and LegacyAge 1.0, and a short shared-screen video of the R-code to show the usage on two example records are accessible on Zenodo (https://doi.org/10.5281/zenodo.5815192; Li et al., 2021b)'.

**Description of code usage in new text (line 331-333):**

**'As for the R-code, users only need to set the working directory where the Bacon results will be stored and input the record ID of interest to run it successfully. The manual and shared-screen video on R-code usage could provide helpful guidance for users, with or without some R-experience'.**

4. **Specific comments**

Reviewer comment: *(1) line 44 - the phrase "calibrated and uncalibrated" is confusing.*

**Response: We rephrased this part of the text. This particular part of the sentence was deleted.**

**New text (line 46-48): 'Furthermore, the chronologies have been established using a variety of methodologies, and the quantification of temporal uncertainty, particularly between records, remains a challenge (Blois et al., 2011; Giesecke et al., 2014; Flantua et al., 2016; Trachsel and Telford, 2017)'.**

Reviewer comment: *(2) line 65-75 - it would be advisable to have these variables in a table with further descriptions.*

**Response: Now, all metadata variables are listed in Supplement Tables S1 and S4 at PANGAEA (https://doi.pangaea.de/10.1594/PANGAEA.933132), the related variable descriptions are listed in the readme text.**

Reviewer comment: *(3) line 79-80 - repeated use of references to "most common"*

**Response: We supplied references where we used most "most common" before.**

**New text (line 84-87): '*Radiocarbon dating:* most records were dated using radiocarbon-based methods ($^{14}$C dating, conventional or accelerator mass spectrometry, Christie, 2018), covering the time range of ca. the last 50 kyr BP (before present, where 'present' is 1950 CE). However, the accuracy and precision of the radiocarbon dates depend on the calibration curve, taphonomy, and dating materials (Blois et al., 2011; Heaton et al., 2021)'.**

Reviewer comment: *(4) Section 2.3.1. - for this type of paper, consider leading this section off with what you have as your final sentence, then describing it. "...all age relationships in our data set are constructed using Bacon..." then describe why and what and how.*

**Response: We followed your suggestion.**

**New text (line 121-128):**

‘We used the Bacon software (Blaauw and Christen, 2011) to establish continuous down-core chronologies from the age control points. Bacon fits a monotonic autoregressive (AR1) model to age control points using Bayesian methods to combine information from the control points with prior information on the statistical properties of accumulation histories for deposits, e.g., a prior distribution for the mean accumulation rate and how it varies (Blaauw and Christen, 2011). Several other approaches are available for age-depth modeling, including linear interpolation, smoothing splines, and other Bayesian methods, e.g., OxCal (Ramsey, 2008) and Bchron (Haslett and Parnell, 2008). However, Bacon has become one of the most frequently used and compares well with other methods (Trachsel and Telford, 2017, Blaauw et al., 2018)’.

Reviewer comment: *(5) line 139-141 - where did the latest calibration curves come from? this sentence lacks context.*

Response: The latest calibration curves (IntCal20, SHcal20, Marine20; http://calib.org/), are already included in Bacon.

**New text (line 148-152):**

‘To transform the measured $^{14}$C ages to calendar ages, the latest calibration curves, approved by the radiocarbon community (Hajdas, 2014), were used in Bacon routine: IntCal20 (Reimer et al., 2020; Heaton et al., 2021) and SHcal20 (Hogg et al., 2020) to calibrate the terrestrial radiocarbon dates in the northern and southern hemispheres, respectively; and Marine20 (Heaton et al., 2020) for the 38 marine records included in our dataset (Sánchez Goñi et al., 2017)’.

Reviewer comment: *(6) Section 2.3.4 consider laying this section out using bullets or with some kind of work design flow infographic.*

Response: We laid this section out using bullets following your suggestion.

**New text (line 161-183):**

‘(1) The prior for the accumulation rate consists of a gamma distribution with two parameters, mean accumulation rate (acc.mean; default 20 yr cm$^{-1}$) and accumulation shape (acc.shape; default 1.5). For the acc.shape, we accepted its default value as higher values resulted in a more peaked shape of the gamma distribution. A first approximation of the acc.mean was calculated as the average accumulation rate between the first and the last date of each record, combined with the prior information of dates, which is more reasonable than using a constant value.

**(2) Bacon divides a core into many vertical sections of equal thickness (thick; default 5 cm), which significantly affects the flexibility of the age-depth model, and through millions of Markov Chain Monte Carlo iterations estimates the accumulation rate for each section. Blaauw and Christen (2011) indicated that models with few sections tend to show more abrupt changes in accumulation rate, while models with many sections usually appear smoother but are computationally more intense. We run Bacon for six section thicknesses (2.5 cm, 5 cm, 10 cm, 30 sections, 60 sections, and 120 sections), optimal values after numerous tests, with and without core-top age resulting in 12 initial chronologies for each record.**

**……'**

Reviewer comment: *(7) * just a note format your units with super- and subscripts, not / notation*

**Response: We changed '/ ' to superscript ($^{-1}$).**

**New text (line 161-162): 'The prior for the accumulation rate consists of a gamma distribution with two parameters, mean accumulation rate (acc.mean; default 20 yr cm$^{-1}$) and accumulation shape (acc.shape; default 1.5)'.**

Reviewer comment: *(8) lines 167 -Consider again bullets or something instead of a numbered list inside of a paragraph.*

**Response: We laid this section out using bullets following your suggestion, same as before.**

**New text (line 186-207):**

**'(1) Reservoir effects: the uptake of old carbon by aquatic plants, mosses, or shells either originating from, e.g., limestone in the catchment ('hard-water effect') or slow $^{14}$C exchange between the atmosphere and ocean interior, can result in too old radiocarbon dates (Philippsen, 2013; Philippsen and Heinemeier, 2013; Giesecke et al., 2014; Heaton et al., 2020). In addition to the reservoir ages reported by the original authors, we also identified some additional records for which there is likely a reservoir effect through modern correction and linear extrapolation (Wang et al., 2017). We then subtracted the reservoir age as a constant from all $^{14}$C dates of an affected record, excluding those derived from terrestrial macrofossils. We may have underestimated the number of such records due to the difficulty of estimating the reservoir age where the sediment surface was eroded or used for agricultural purposes.**

**(2) Waterline issues: stratigraphic records do not always start at a depth of 0 cm, for example, if the uppermost part of the core is lost, if the record is only a part of a longer sequence, or if the depths are measured from the water surface instead of the sediment surface, leading to the so-**

**called waterline issue. Accordingly, we adjusted the uppermost depth of the chronology based on information collected from the original publications and Neotoma.**

**……'**

---

## Author Comment (AC5)

1. **General comments**

   Reviewer comment: Most analyses using Neotoma or other archived pollen data are dependent, at least to some extent, on the chronologies. The available chronologies have variable quality: some record have an uncalibrated chronology, others have a Bayesian chronology. In many cases the uncertainty on the chronology is not available, or if it is, just the upper and lower credibility interval. To synthesise pollen data from several datasets, it may be necessary to harmonised the age-depth models, a huge amount of work. Once such harmonisation is presented in this current manuscript.

   Reviewer comment: *(1) As far as I can tell, the chronologies are not archived, but instead the metadata needed to make the chronologies. This is probably a good idea as it encourages the user to check the parameters.*

   **Response: We are sorry for the misunderstanding. We provided the final chronologies as well as all metadata and scripts to recreate the chronologies. We also improved the documentation on accessing and using the dataset and code to avoid further misunderstandings.**
   **New text (line 312-333):**
   **'4 Code and data availability**
   **Seven supplementary datasets (Table S1-S7, in comma-separated values format) and one readme text about the LegacyAge 1.0 are accessible in the navigation bar 'Further details' of the PANGAEA page (https://doi.pangaea.de/10.1594/PANGAEA.933132; Li et al., 2021a). We provided the chronological control points metadata (Table S1), prior information of dates from publication (Table S2), Bacon parameter settings (Table S3), original chronology metadata from the Neotoma and Cao et al. (2013, 2020) (Table S4), LegacyAge 1.0 chronology (Table S5), description of the comparison of original chronology and LegacyAge 1.0 (Table S6), and record references (Table S7) respectively. All datasets are already in long data format that can be joined by the dataset ID.**

   **The R-code for calculation and comparison of chronologies with embedded manual, metadata for code runs, Bacon output graphs of each record, graphs comparison of original chronologies and LegacyAge 1.0, and a short shared-screen video of the R-code to show the usage on two example records are accessible on Zenodo (https://doi.org/10.5281/zenodo.5815192; Li et al., 2021b).**

   **5 How to use the LegacyAge 1.0 dataset and code**

   **LegacyAge 1.0 provides the calibrated ages (mean, median, minimum, maximum) and uncertainties at each centimeter for each record with a 95% confidence interval (Supplement**

**Table S5). All users can apply some interpolation algorithms in the chronologies, subsetted from the LegacyAge 1.0 dataset or outputted by our code, to assign ages for proxy depths of records.**

**As for the R-code, users only need to set the working directory where the Bacon results will be stored and input the record ID of interest to run it successfully. The manual and shared-screen video on R-code usage could provide helpful guidance for users, with or without some R-experience'.**

Reviewer comment: *(2) One important result is that "95.4% of records could be improved ". However, it is unclear what objective criteria were used to determine whether the new model represented an improvement. The criteria need to be explicitly stated.*

**Response: We added the criteria for the preferred models in the new text.**

**New text (line 232-237): 'We plotted our newly generated 'best' calibrated chronologies with 95% confidence intervals together with the original ones taken from the Neotoma and Cao et al. (2013, 2020) datasets (Supplement Table S4) to compare and evaluate the performance of the new models visually. The criteria for the preferred models are that the model fitted the dates well, had small uncertainties, combined dates with prior information (e.g., geological and hydrological setting, environmental history), and calibrated with the latest calibration curves'.**

Reviewer comment: *(3) The metadata and code are available on github (Zenodo.org would be preferable for the final version).*

**Response: We agree with your suggestion. Therefore, we moved to store metadata, code, and results from GitHub to Zenodo (https://doi.org/10.5281/zenodo.5815192).**

2. **Data (PANGAEA)**

Reviewer comment: *(1) The data are arranged in wide format, with a set of columns for each date. This is not the ideal way to arrange the data, as it makes the code much more complicated to deal with this structure, and will need extra extra columns adding in the future to cope with new sites. A better setup would be to have the data in long format, perhaps in multiple files that can be joined by the dataset ID.*

**Response: All datasets are already in long data format that can be joined by the dataset ID. To avoid misunderstanding, we provide this information now in the text.**

**New text (line 319-320): 'All datasets are already in long data format that can be joined by the dataset ID'.**

Reviewer comment: *(2) At present, datasets are marked as being marine or otherwise. At least in principle, there could be datasets where some dates are on marine fossils, and others on terrestrial macrofossils. Marine should be a property of the date, not the core.*

**Response: Thank you for this comment. After rechecking all sites marked as marine in the dataset, we did not find a single case where terrestrial material was dated. So we did not implement this suggestion.**

Reviewer comment: *(3) Setting a constant reservoir age for a core without error seems risky to me. Do you check these are only occuring with bulk dates, as there should not normally be a problem with terrestrial macrofossils.*

**Response: Although the reservoir ages with errors would be better, the reservoir ages reported by the original author through various methods are mostly without error. We fully respect the original authors' comments because we assume that they are more familiar with the sites than we are. As you can infer from the variable 'Material_Dated' in supplement Table S1, various dating materials were used for dating. Thus, we subtracted the reservoir age as a constant from all $^{14}$C dates of an affected record, excluding those derived from terrestrial macrofossils.**

**New text (line 187-193):**

**'Reservoir effects: the uptake of old carbon by aquatic plants, mosses, or shells either originating from, e.g., limestone in the catchment ('hard-water effect') or slow $^{14}$C exchange between the atmosphere and ocean interior, can result in too old radiocarbon dates (Philippsen, 2013; Philippsen and Heinemeier, 2013; Giesecke et al., 2014; Heaton et al., 2020). In addition to the reservoir ages reported by the original authors, we also identified some additional records for which there is likely a reservoir effect through modern correction and linear extrapolation (Wang et al., 2017). We then subtracted the reservoir age as a constant from all $^{14}$C dates of an affected record, excluding those derived from terrestrial macrofossils'.**

**3. Code**

Reviewer comment: *(1) The code is presented in a single script. This is fine if someone wants to make chronologies for all datasets, but often people will want to run a subset of the sites, and may find the script difficult to use. Breaking the script into functions than each do one task, perhaps compiled into a small package with help files and other documentation, would make the product more useful.*

**Response: We apologize for this. We revised the code and rephrased this part of the text.**

**Description of the uploaded file in new text (line 321-324):**

'The R-code for calculation and comparison of chronologies with embedded manual, metadata for code runs, Bacon output graphs of each record, graphs comparison of original chronologies and LegacyAge 1.0, and a short shared-screen video of the R-code to show the usage on two example records are accessible on Zenodo (https://doi.org/10.5281/zenodo.5815192; Li et al., 2021b)'.

**Description of code usage in new text (line 331-333):**

'As for the R-code, users only need to set the working directory where the Bacon results will be stored and input the record ID of interest to run it successfully. The manual and shared-screen video on R-code usage could provide helpful guidance for users, with or without some R-experience'.

Splitting the code into separate functions is an interesting idea, but getting this to universal usage (now it is very specific and focused on our data) would need more modifications. But to consider this, we separated the "Age Allocation" part into a standalone code (line 252-295 of code) that should be the most valuable part of the main script for other purposes.

**New code (line 252-295):**

'#-----Age Allocation-----

 # Define basic values for the age allocation

 model.AWI <- read.table(paste0(folder, "/Ages.txt/", ID, ".txt"), header = TRUE)

model.rest <- round(model.AWI$depth[length(model.AWI$depth)] - floor(model.AWI$depth[length(model.AWI$depth)]), 4)

 new.memory <- data.frame(min = NA, max = NA, median = NA, mean = NA, stringsAsFactors = FALSE)

......'.

Reviewer comment: *(2) The code includes fixes for issues the authors found in Neotoma. Please report these issues, and make sure the code to fix them (e.g. line 187) will work safely when the error is fixed.*

**Response: We discovered wrong depth units of the four IDs (15669, 15671, 15673, 156750) reported it to the Neotoma team, which revised them. So this problem does not anymore apply.**

**4. Figure**

Reviewer comment: *(1) Figures 4-6. The x-axis, labelled 0-1, 1-2 etc, is rather cluttered and difficult to read. Please consider other ways to label them. The relative lack of outliers in the pre-LGM sediment is interesting, but probably represents over-fitting the models to sparse data.*

**Response: We replotted figures.**

**New Figures:**

[Figure]

**Figure 4. Histogram showing the number of available dates in distinct time slices.**

[Figure]

**Figure 5. Histogram showing the number of available chronologies in distinct time slices.**

[Figure]

**Figure 6. Boxplots of age uncertainties and outlier percentages in distinct time slices.**

As we can see from the figure below (Figure R1), the outliers appear on the maxima side of the boxplot. After examining these outliers, we found that most of them came from chronologies with sparse age control points and significant dating errors. For example, the maxima values in all distinct time slices are from the *Nachtigall* record (Dataset ID 41435; Figure R2), with only three significant-error dates.

New text (line 280-282): **'The boxplots show wide boxes, i.e., a more extensive data range, for the LGM period, characterized by fewer outliers, mostly from chronologies with sparse age control points and significant dating errors, than the periods with small box sizes'.**

[Figure]

**Figure R1. Boxplots of age uncertainties and outlier percentages in distinct time slices (with outliers).**

[Figure]

**Figure R2. Bacon output graph of the *Nachtigall* record (Dataset ID 41435).**

Reviewer comment: *(2) Figure 7 has a minuscule font: I need to view it at 200% before I can read it. No need to repeat the legend four time - once is enough and leaves more space for everything else.*

**Response: We laid out this figure on a whole page. As suggested by reviewer #3, we have newly provided the section thicknesses in the title of each figure. Also, one additional age-depth model created by Bacon with relatively poor performance has been included for comparison.**

**5. Specific comments**

Reviewer comment: *Line 202 "with fewer than 2" Maybe rewrite as "only one" for clarity. Consideration should be given as to the minimal number of dates that can give a good chronology - I would be cautious using a model based on only two dates.*

**Response: Thanks for your suggestion. We rephrased this part of the text to avoid further misunderstandings.**

**New text (line 241-243): 'We discarded 640 records with fewer than two reliable dates (i.e., no reliable date or only one reliable date), evaluated based on prior information from original literature, leaving chronologies for 2831 records'.**

**It is well known that the quality of the chronology is closely related to the quality and quantity of the date. As a rule, the more high-quality dates a chronology has, the more reliable it is.**

Therefore, it is difficult to give the minimum number of dates to establish a chronology. As you read from column *'Dates_Number'* of the supplement Table S5, 9.7% of the records have only two dates. We have kept these to keep as many records as possible.

---

## Author Comment (AC6)

**1. General comments**

Reviewer comment: In this work the authors present their attempt to harmonize mainly radiocarbon-based chronologies of continental climate records. The harmonisation is with respect of age-model software usage, calibration curve usage, which is a very valuable task. Furthermore, harmonisation is performed with respect to parameters used for the age-depth modelling software. As far as I understand, the authors use the age-modelling software Bacon for age-depth modelling of a huge quantity of records. Before modelling, the cores were manually evaluated in terms of complications, such as radiocarbon reservoir effects, water lines, etc.

 While I appreciate their approach, I think there are some things to be improved before suggesting this piece of work for publication.

**2. Data (PANGAEA)**

Reviewer comment: *(1) Furthermore, I am not able to find age-depth profiles on their provided Pangaea-page. I thought the authors did all their work (handling reservoir effects, water lines, deciding for the best thicknesses to be applied, …) in order to provide a homogeneous age-depth data set. And according to their paper, they spend a lot of efforts to evaluate the datings etc of all records. It would be a pity, if they would not share this. Or is the user supposed to start from scratch again? Even if it 'only' means to run their script – if I understand the code structure correctly, the user has to run all of their thousands of records, even if the user is only interested in one or two records. Especially, as this means to run 'millions of MCMC iterations' (line 120) which cannot be that cheap as even admitted by the authors: "… it needs much supervision and computing power" (line 122). Why not provide all age depth models (including uncertainties) in addition to all meta data and code?*

**Response: Thanks for your suggestion. We provided all age-depth models (including uncertainties) on PANGAEA (https://doi.pangaea.de/10.1594/PANGAEA.933132; Supplement Table S3) and improved the documentation on the files uploaded in the text.**

**New text (line 312-324):**

**'4 Code and data availability**
**Seven supplementary datasets (Table S1-S7, in comma-separated values format) and one readme text about the LegacyAge 1.0 are accessible in the navigation bar 'Further details' of the PANGAEA page (https://doi.pangaea.de/10.1594/PANGAEA.933132; Li et al., 2021a). We**

**provided the chronological control points metadata (Table S1), prior information of dates from publication (Table S2), Bacon parameter settings (Table S3), original chronology metadata from the Neotoma and Cao et al. (2013, 2020) (Table S4), LegacyAge 1.0 chronology (Table S5), description of the comparison of original chronology and LegacyAge 1.0 (Table S6), and record references (Table S7) respectively. All datasets are already in long data format that can be joined by the dataset ID.**

**The R-code for calculation and comparison of chronologies with embedded manual, metadata for code runs, Bacon output graphs of each record, graphs comparison of original chronologies and LegacyAge 1.0, and a short shared-screen video of the R-code to show the usage on two example records are accessible on Zenodo (https://doi.org/10.5281/zenodo.5815192; Li et al., 2021b)'.**

Reviewer comment: *(2) Another critical question is about the final age models. As I cannot find them, nor are able to run the R script, I have to ask: Which depths intervals do you choose to save for the homogenised age-depth models? In the paper you mention the effect of choosing different levels or depth intervals on the goodness of the model data and that some are better suited than others. However, I even wonder, why a user should care about having the age-depth relationship on a fixed sampling interval? If I want to work with other paleoclimate data, I am interested in an age-depth model, which provides dates at depth, where the proxies were measured. Unfortunately, this is not mentioned in the paper. Or do you expect the user to apply some (more or less) fancy interpolation algorithm to assign ages for the proxy depths?*

**Response: We chose a depth interval of 1 cm to save for the harmonized age-depth models. We reorganized the description of the two Bacon parameters ('thick' and 'd.by') and apologize for the confusion.**

**'thick' description in the new text (line 167-173):**

**'(2) Bacon divides a core into many vertical sections of equal thickness (thick; default 5 cm), which significantly affects the flexibility of the age-depth model, and through millions of Markov Chain Monte Carlo iterations estimates the accumulation rate for each section. Blaauw and Christen (2011) indicated that models with few sections tend to show more abrupt changes in accumulation rate, while models with many sections usually appear smoother but are computationally more intense. We run Bacon for six section thicknesses (2.5 cm, 5 cm, 10 cm, 30 sections, 60 sections, and 120 sections), optimal values after numerous tests, with and without core-top age resulting in 12 initial chronologies for each record'.**

**'d.by' description in the new text (line 182-183):**

**'The parameter 'd.by' (default 1 cm) defines the depth intervals at which ages are calculated, and we accepted its default value'.**

We added a new section in the text to introduce how to assign ages for the proxy depths. For example, we are applying linear interpolation to assign the ages of pollen samples for those records.

New text (line 327-330):

**'5 How to use the LegacyAge 1.0 dataset and code**

**LegacyAge 1.0 provides the calibrated ages (mean, median, minimum, maximum) and uncertainties at each centimeter for each record with a 95% confidence interval (Supplement Table S5). All users can apply some interpolation algorithms in the chronologies, subsetted from the LegacyAge 1.0 dataset or outputted by our code, to assign ages for proxy depths of records'.**

**3. Code**

Reviewer comment: *(1) Usually, such a data set and code is generated to be used. Unfortunately, I cannot find any description or manual, how to access the age-depth models. Nor is it possible for me to run the R-script. I admit, I am a R-noob, but I think, application should be properly described with at least a short manual for users with some R-experience (or even noobs). This does not have to come with this publication, but it should at least appear on their github space next to the R-file.*

*Or at least enable the user to only calculate the age-depth models of the records they are interested in?*

**Response: We apologize for this. We revised the code and provided the manual and shared-screen video on R-code usage. We also added a description in the text to introduce how to use the code.**

**New text (line 331-333):**

**'As for the R-code, users only need to set the working directory where the Bacon results will be stored and input the record ID of interest to run it successfully. The manual and shared-screen video on R-code usage could provide helpful guidance for users, with or without some R-experience'.**

Reviewer comment: *(2) Is the output of your script arranged in a way, that this could be easily accessed?*

**Response: We revised the code and provided a manual inside the code. Our code automatically places the different types of files outputted by Bacon in different folders, which will help the users quickly find the files they need.**

**Manual in the code (line 11-19):**

**'#-----Resultfolders----------------------------------**
**# Ages.txt       ->  Chronology tables by Bacon**
**# Bacon.pdf      ->  Outputplot by Bacon**
**# Calibration    ->  Plots from Calibration**
**# ID.Subsets     ->  Summarized data of the ID**
**# Plot.png        ->  Plot to compare with other chronologies**
**# Plot.flipped   ->  the same plot but flipped**
**# Sites           ->  all data concerning the ID'.**

**4. Figure**

Reviewer comment: *Fig. 7: Please provide information about which of the twelve generated age-depth models for each record you show here! Would it be possible to show one additional age-depth realisation, which fits less good with the measured ages. Only to give the reader an idea about the effects of the choice of depths intervals.*

**Response: Thanks for your suggestion. We laid out this figure on a whole page, so we can't show it here. We provided the section thicknesses in the title of each figure and added one additional age-depth realization established by Bacon, which fit less well with the measured ages.**

**New text (line 296-297):**

**'Selected typical examples of the comparative results between the accepted LegacyAge 1.0 chronologies, alternative newly generated but rejected chronologies, and the original chronologies are illustrated in Fig. 7'.**

**Title of Figure 7 in the new text (line 311):**

**'Figure 7. Comparison of LegacyAge 1.0 chronologies with the original ones. Green line: original chronology. Blue line: LegacyAge 1.0 chronology. Yellow line: alternative newly generated but rejected chronology. Red: date in chronology metadata. Pink: date from prior information. Grey shading: age uncertainties (95% confidence'.**

**5. Specific comments**

Reviewer comment: *(1) L16 and 46: Please elaborate a bit more on what you understand by 'harmonized chronology' already this early in the manuscript. I am pretty, sure, that different people understand different things under this term. I mean later in the paper it becomes clear, what you understand by this term, but I think it is worth to highlight this already in the beginning of your work.*

**Response: According to your suggestion, we elaborated this term a bit more in the introduction section.**

**New text (line 48-51): 'Recently, the need for harmonized and consistent chronologies allowing for the accurate assessment of temporal uncertainty between records has increased as studies are looking for spatiotemporal patterns using multi-record analyses (Jennerjahn et al., 2004; Blaauw et al., 2007; Giesecke et al., 2011; Flantua et al., 2016)'.**

Reviewer comment: *(2) L27-28: This sentence needs more explanations. Maybe not here in the abstract, but below in the according text passages. Please find a more detailed comment below.*

**Response: Yes, only the final result of the comparison is shown here. We list the criteria below.**

**New text (line 234-237): 'The criteria for the preferred models are that the model fitted the dates well, had small uncertainties, combined dates with prior information (e.g., geological and hydrological setting, environmental history), and calibrated with the latest calibration curves'.**

Reviewer comment: *(3) L69-74: You provide quite some detailed information on metadata, which I appreciate a lot. However, I doubt that putting those data all in one file is the best option. I agree with referee 2 to splitting this file up in several is maybe more appropriate and easier to handle. At least keep this in mind for any potential future improvements.*

**Response: We provided seven supplementary datasets in long data format that can be joined by the dataset ID. To avoid misunderstanding, we provide this information now in the text.**

**New text (line 313-320):**

**'Seven supplementary datasets (Table S1-S7, in comma-separated values format) and one readme text about the LegacyAge 1.0 are accessible in the navigation bar 'Further details' of the PANGAEA page (https://doi.pangaea.de/10.1594/PANGAEA.933132; Li et al., 2021a). We provided the chronological control points metadata (Table S1), prior information of dates from publication (Table S2), Bacon parameter settings (Table S3), original chronology metadata from the Neotoma and Cao et al. (2013, 2020) (Table S4), LegacyAge 1.0 chronology (Table S5),**

**description of the comparison of original chronology and LegacyAge 1.0 (Table S6), and record references (Table S7) respectively. All datasets are already in long data format that can be joined by the dataset ID'.**

Reviewer comment: *(4) L155: 'acc.mean' is possibly 'acc.rate'?*

**Response: The correct abbreviation for mean accumulation rate is 'acc.mean'. We have made the change in the text and apologize for the confusion.**

**New text (line 161-162): 'The prior for the accumulation rate consists of a gamma distribution with two parameters, mean accumulation rate (acc.mean; default 20 yr cm$^{-1}$) and accumulation shape (acc.shape; default 1.5)'.**

Reviewer comment: *(5) L158: 'We tested six thicknesses (2.5 cm, 5 cm, 10 cm, 30 sections, 60 sections, and 120 sections) ...'. I am not very familiar with Bacon. But, why would you want to test those 6 sampling intervals? I mean, the proxies of the cores were measured at specific depths - wouldn't it be more suitable to only interpolate to those depths, where proxy data exist? Actually, this is the data, I would be interested in. But it seems, that this is missing completely. What do you suggest to finally obtain the ages at those depths?*

**Response: The selected section thicknesses (2.5 cm, 5 cm, 10 cm, 30 sections, 60 sections, and 120 sections) are the optimal values after numerous tests.**

**We added a new section in the text to introduce how to assign ages for the proxy depths.**

**New text (line 326-330):**

**'5 How to use the LegacyAge 1.0 dataset and code**

**LegacyAge 1.0 provides the calibrated ages (mean, median, minimum, maximum) and uncertainties at each centimeter for each record with a 95% confidence interval (Supplement Table S5). All users can apply some interpolation algorithms in the chronologies, subsetted from the LegacyAge 1.0 dataset or outputted by our code, to assign ages for proxy depths of records'.**

Reviewer comment: *(6) L159: 'artificial surface age', Why would it be necessary to add an artificial date? I don't know if I understand the concept of adding an artificial date correctly. Stating things like this sounds very arbitrary. Or do you mean you added another age-constraint due to the assumption that the core sedimentation was active until core recovery? And that the additional age constraint is the year of core recovery? If yes, please consider to specify accordingly.*

**Response: Yes, you are right. To avoid misunderstanding, we rephrased this part of the text.**

**New text (line 139-143):**

'For modern core-tops, if the core was collected from sites where sediment was still accumulating, the sediment surface could be assigned to the year of sampling, adding one significant time control for the chronologies. If the sampling date was unavailable, an alternative surface age from the original chronology in Neotoma was added at the core top. An estimated artificial core-top age (-50 + -30 cal yr BP) was used if none of the above ages were available (Supplement Table S2, S3)'.

Reviewer comment: *(7) L159: 'generating 12 age models for each core'. Just to make sure I understand correctly. Your code provides 12 age-depth models for one core. Are all provided in output files?*

**Response: Yes, our code initially outputs 12 age-depth models for each record. We only provided the parameter settings of the 'best' chronology for each record. You will get the best chronology for each record if you run the script directly. Meanwhile, if you want to get multiple age-depth models for each record, you can do so by modifying the column *'Resolution.cm'* or *'Resolution.section'* of Table S3.**

Reviewer comment: *(8) L170: I think, C exchange between dissolved C-species in water and atmospheric CO2 is not responsible for 'too old radiocarbon dates'. Instead, this process counter balances to some degree the effect of the arguments listed earlier in this sentence.*

**Response: We agree. We rephrased this part of the text to avoid further misunderstandings.**

**New text (line 187-190):**

'Reservoir effects: the uptake of old carbon by aquatic plants, mosses, or shells either originating from, e.g., limestone in the catchment ('hard-water effect') or slow $^{14}$C exchange between the atmosphere and ocean interior, can result in too old radiocarbon dates (Philippsen, 2013; Philippsen and Heinemeier, 2013; Giesecke et al., 2014; Heaton et al., 2020)'.

Reviewer comment: *(9) L171-173: For some records you added your evaluation of reservoir effects. I appreciate this a lot, but I think it is worth to add a column in your metadata file and mark those records. This would allow a better transparency about what is your evaluation and which information came from the original studies.*

**Response: We listed the reservoir age in column *'Reservoir'* of supplement table S3. Readers can also view 'type 2' (Record with reservoir effect) in column *'Category'* of supplement table S2 to learn how obtained the reservoir effect.**

Reviewer comment: *(10) L184: For the use of radiocarbon dates for modelling purposes, you followed 'in most cases the suggestions in the original publications'. Please consider – again for a better transparency - to provide information (maybe in your metadata file), for which records you did not follow the suggestions of the original publications.*

**Response: Thanks for your suggestion. We listed all prior information collected from the original publication in supplement Table S2. To avoid misunderstanding, we provided an example in the text.**

**New text (line 207-209): 'For example, we excluded the date at 164 cm, accepted by the author (Gajewski et al., 2000), from the *Muskoka Lake* record (ID 1783), as it does not agree with the other three dates from the same core and where lithology had changed significantly at that depth'.**

Reviewer comment: *(11) L189-191: 'For each record, 12 age models were visually assessed. Preference was given to models that fitted the dates well and with small uncertainties when choosing the 'best' model for each record (Blaauw and Christen, 2011; Blaauw et al., 2018).'. This is a lot of work for thousands of records. You are sure, that you did this all correctly for this large amount of records? I wonder if it would have been more objective to apply a short statistical test on this. I mean, most likely a simple least square test between age model and ages of dated depths would do a better and faster job. Also the 'small uncertainty' argument would be most likely more precise and faster to obtain, when calculating the mean uncertainty instead relying on visual assessment.*

**Response: We rephrased this part of the text according to your suggestion.**

**New text (line 214-227):**

**'To objectively evaluate the 12 initial age-depth models for each record, we initially tested a least-squares method between the age model and ages of dated depths and calculated the mean uncertainty for each model. However, the least-squares method is susceptible to outliers (Birks et al., 2012), and models with least-squares may risk more abrupt changes in accumulation rate due to over-fitting dates. Instead of a numerical comparison, we finally implemented a visual comparison based on the Bacon output graphs, which show the Markov Chain Monte Carlo iterations, the prior and posterior distributions for the accumulation rate and memory, and how well the model fits the date (Blaauw and Christen, 2011).**

**Preference was given to models that fitted the dates well, had small mean uncertainties (Supplement Table S5), and good runs of Markov Chain Monte Carlo iterations (i.e., a stationary distribution with little structure among neighboring iterations as indicated by the traceplot of the joint likelihood) when visual choosing the 'best' model for each record (Blaauw and Christen,**

**2011; Blaauw et al., 2018). If necessary, we adjusted the parameter settings such as the section thickness and mean accumulation rate to better fit with the dates that were consistent with prior information. For the final parameter settings used for each record, please see https://doi.pangaea.de/10.1594/PANGAEA.933132 (Supplement Table S3; Li et al., 2021a)'.**

Reviewer comment: *(12) L203: Who did the evaluation about what a reliable date is? You or the original authors? I can imagine, that this is a difficult task, especially for cores from others.*

**Response: Sorry for the confusion; we rephrased this part of the text.**

**New text (line 204-209):**

**'Dates rejected/added: Neotoma usually reports all [14]C dates from cores, even when deemed inaccurate. We assessed prior information on dates and then excluded the [14]C dates of samples with contaminated or reworked sediments from age-depth model from age-depth models, in most cases following the suggestions in the original publications. For example, we excluded the date at 164 cm, accepted by the author (Gajewski et al., 2000), from the *Muskoka Lake* record (ID 1783), as it does not agree with the other three dates from the same core and where lithology had changed significantly at that depth'.**

Reviewer comment: *(13) L247-248: 'where original chronologies outperformed LegacyAge 1.0, ...' How do you know, which model approach outperforms the other? How can you measure or evaluate this? Do you have knowledge of the 'true sedimentation history' of all those records to be able to judge this? Which one do you choose from your 12 ones/core? I think it is very crucial to provide more details on this issue. Or, in case you wanted to express a different thing with this expression, please consider to rephrase this sentence.*

**Response: Yes, you are right. We provided the criteria for comparison and rephrased this part of the text.**

**New text (line 234-237):**

**'We plotted our newly generated 'best' calibrated chronologies with 95% confidence intervals together with the original ones taken from the Neotoma and Cao et al. (2013, 2020) datasets (Supplement Table S4) to compare and evaluate the performance of the new models visually. The criteria for the preferred models are that the model fitted the dates well, had small uncertainties, combined dates with prior information (e.g., geological and hydrological setting, environmental history), and calibrated with the latest calibration curves'.**

**New text (line 286-291):**

'For 906 records out of the 2831 records included in the LegacyAge 1.0, no calibrated chronologies were originally available from the Neotoma and Cao et al. (2013, 2020) datasets for comparison. Of the remaining 1925 records, the new LegacyAge 1.0 chronologies were selected instead of the original ones in 95.4% of cases, based on the aforementioned criteria. However, some records still chose the original chronology, mainly because they are varve chronologies, had incomplete metadata (e.g., missing sample depths), or included some non-[14]C dates that our model could not accommodate (Supplement Table S6)'.